# Training Dynamics of Learning 3D-Rotational Equivariance

**Max W. Shen**[*]                                                                 *shenm19@gene.com*
*Genentech Computational Sciences*

**Ewa M. Nowara**[*]
*Genentech Computational Sciences*

**Michael Maser**
*Genentech Computational Sciences*

**Kyunghyun Cho**
*Genentech Computational Sciences & New York University*

**Reviewed on OpenReview:** *https:// openreview. net/ forum? id=DLOIAW18W3*

## Abstract

While data augmentation is widely used to train symmetry-agnostic models, it remains unclear how quickly and effectively they learn to respect symmetries. We investigate this by deriving a principled measure of equivariance error that, for convex losses, calculates the percent of total loss attributable to imperfections in learned symmetry. We focus our empirical investigation to 3D-rotation equivariance on high-dimensional molecular tasks (flow matching, force field prediction, denoising voxels) and find that models reduce equivariance error quickly to $\leq 2\%$ held-out loss within 1k-10k training steps, a result robust to model and dataset size. This happens because learning 3D-rotational equivariance is an easier learning task, with a smoother and better-conditioned loss landscape, than the main prediction task. For 3D rotations, the loss penalty for non-equivariant models is small throughout training, so they may achieve lower test loss than equivariant models per GPU-hour unless the equivariant "efficiency gap" is narrowed. We also experimentally and theoretically investigate the relationships between relative equivariance error, learning gradients, and model parameters.

## 1 Introduction

Machine learning modeling of molecules – generative modeling, property prediction, simulating dynamics, etc. – holds great potential for advancing scientific discovery and human health via therapeutics. Molecules are three-dimensional physical entities whose biochemical properties are invariant or equivariant to 3D rotations[1]. To model these symmetries, two approaches are common: 1) use symmetry-respecting neural architectures, or 2) training symmetry-agnostic models with data augmentation, wherein training samples are randomly transformed by the symmetry group. This choice is made at the start of any molecular modeling project and can have a significant impact on engineering, training, and model performance, yet there has been a lack of clarity on when to prefer which approach.

3D-rotational equivariant architectures use sophisticated tensor operations to maintain equivariance (Luo et al., 2024), achieve loss scaling curves similar to non-equivariant models (Brehmer et al., 2025; Mahan et al., 2024), and are more parameter efficient than non-equivariant models on spherical image tasks (Gerken et al., 2022). Yet they can be much slower (10x-100x) than non-equivariant models[2] (Gerken et al., 2022; Elhag et al., 2025; Brehmer et al., 2025), and they can be harder to optimize based on findings that breaking exact

---

[*]Co-first author.
[1]Molecules also have symmetries to translation, which are commonly handled by centering molecule positions.
[2]Slowness is also partially from less optimized code and GPU kernels.

equivariance improves learning (Qu & Krishnapriyan, 2024; Pertigkiozoglou et al., 2024; Canez et al., 2024). We call this the *efficiency gap*, arising both from optimization speed (training steps per second) and ease (loss reduction per training step). Meanwhile, recent work achieve strong performance on molecular machine learning tasks using non-equivariant architectures with data augmentation (Wang et al., 2024; Abramson et al., 2024; Qu & Krishnapriyan, 2024; Geffner et al., 2025).

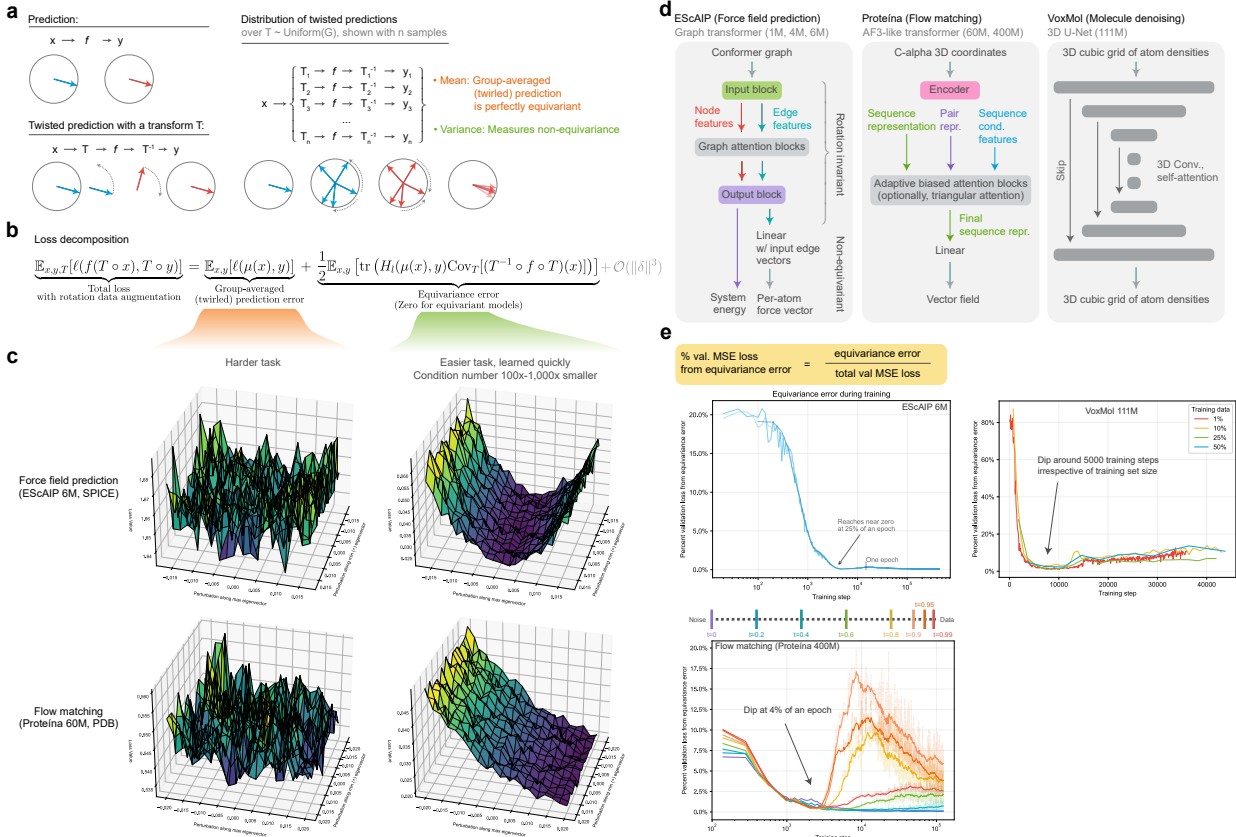

Figure 1: Overview of the paper. (a) Schematic of twisting and twirling, which underpin a principled measure of equivariance error. (b) Loss decomposition by Taylor expansion around the twirled prediction. (c) Loss landscapes for each loss component at early model checkpoints (step=500). (d) Architectures of three non-equivariant models studied here. (e) For MSE loss, the loss decomposition holds exactly, enabling computing the percent validation loss from equivariance error, which is plotted by training step in three settings.

To answer "are symmetry-respecting architectures worth it?", one powerful principle is: *use the model that achieves better held-out loss.* In a fixed amount of GPU-hours, non-equivariant models could incur "unnecessary" equivariance error leading to higher loss, but equivariant models may achieve worse test loss due to the efficiency gap. In fact, we suggest the loss penalty vs. efficiency gap tradeoff is a general explanatory framework. This work focuses on equivariance, because on rotation-invariant tasks like property prediction, symmetry-respecting architectures are relatively uncontroversial (Shoghi et al., 2024; Gasteiger et al., 2022; Nowara et al., 2025): they have a minimal efficiency gap to symmetry-agnostic architectures as rotation-invariant features are informative and fast to compute, and standard deep learning operations easily preserve rotation invariance. In contrast, consider set permutation invariance where the symmetry-respecting architecture is the norm. This can be explained by observing that set transformers have minimal efficiency gap to symmetry-agnostic transformers, as set transformers simply ignore positional embeddings.

While it is possible to directly compare efficiency gaps to loss penalties from imperfect symmetry, this is easily confounded by implementation details. To provide a more fundamental insight, we instead isolate and quantify a key source of potential underperformance in symmetry-agnostic models. We develop tools to investigate: *what is the percent of a symmetry-agnostic model's loss that comes only from its failure*

*to be perfectly equivariant?* (§2, Fig. 1A-B) In an idealized setting (ignoring efficiency differences), this characterizes the counterfactual error reduction if we had trained a symmetry-respecting model instead. In light of efficiency gaps for 3D-rotational equivariance, this metric quantifies how small the efficiency gap must become for equivariant models to outperform non-equivariant models.

In this work, we focus our empirical investigations to three high-dimensional ($\mathbb{R}^{3N} \to \mathbb{R}^{3N}$) molecular learning tasks satisfying 3D-rotational equivariance – flow matching, molecular dynamics force field prediction, and denoising voxelized atomic densities (§3, Fig. 1D). For any non-negative convex loss, we decompose the total loss with data augmentation $\mathcal{L}(\theta) = \mathcal{L}_{\text{mean}}(\theta) + \mathcal{L}_{\text{equiv}}(\theta)$, where $\mathcal{L}_{\text{equiv}}$ captures all information about deviance from exact equivariance. In particular, exactly equivariant models have $\mathcal{L}(\theta) = \mathcal{L}_{\text{mean}}(\theta)$, i.e., $\mathcal{L}_{\text{equiv}} = 0$.

We find that equivariance error shrinks rapidly within 1k-10k training steps (minutes) to under 2% of the total loss (Fig. 1E). This occurs because $\mathcal{L}_{\text{equiv}}$ is a significantly easier learning task than $\mathcal{L}_{\text{mean}}$: the loss landscape for $\mathcal{L}_{\text{equiv}}$ is significantly smoother and better conditioned (Fig. 1C). Strikingly, this is robust to model size, training set size, batch size, and optimizer: we find it with standard batch sizes as well as batch size 1, on training sets of 1M molecules to as small as 500 molecules, and on model sizes from 1M to 400M.

Lastly, in §4, we conduct theoretical and experimental investigations to better understand the relationship between relative equivariance error, learning gradients, and parameters. We prove that under certain conditions, and experimentally, smaller equivariance error can increase the similarity between $\mathcal{L}(\theta) \approx \mathcal{L}_{\text{mean}}(\theta)$. We also prove a quadratic relationship between $\mathcal{L}_{\text{equiv}}$ and the parameter deviation from the subspace of exactly equivariant functions for the modern graph transformer EScAIP.

## 2    Measuring Equivariance & Loss Decompositions

Let $f : \mathbb{R}^D \to \mathbb{R}^D$ be a learnable function and let $G$ be a compact group, for instance of 3D rotations. We consider $T$ as the matrix representation of the action of $G$ on $\mathbb{R}^D$. A function $f$ is $G$-equivariant if it commutes with all transformations $T \in G$, such that for any input $x \in \mathbb{R}^D$, we have $f(T(x)) = T(f(x))$, also written $(f \circ T)(x) = (T \circ f)(x)$. Rearranging, we observe that a perfectly equivariant function satisfies, for all $x, T$:

$$(T^{-1} \circ f \circ T)(x) = f(x) \tag{1}$$

We call $(T^{-1} \circ f \circ T)(x)$ the *twisted prediction* for $x$, from the *twisted function* $T^{-1} \circ f \circ T$. To produce a twisted prediction[3] on molecules, we sample a random rotation, use it to rotate the input molecule, pass this through the function, and un-rotate the output. The un-rotation step re-aligns the output to the "original frame" of the input molecule, which provides a canonical frame to compare the impact of different transformations on the output.

In contrast to a perfectly equivariant function, a non-equivariant function must have some distinct transformations $T_1, T_2$ where the twisted prediction is different: $(T_1^{-1} \circ f \circ T_1)(x) \neq (T_2^{-1} \circ f \circ T_2)(x)$. This property motivates analyzing the distribution of twisted predictions over a uniform distribution on the group, which is the usual choice for data augmentation. For a given $x$:

$$Z_x(T) \triangleq (T^{-1} \circ f \circ T)(x), \quad T \sim \text{Uniform}(G) \tag{2}$$

Its first central moment $\mu(x)$ is the group-averaged, or *twirled* prediction.

$$\mu(x) \triangleq \mathbb{E}_T[(T^{-1} \circ f \circ T)(x)] \tag{3}$$

---

[3]The name reflects a physical intuition of introducing a twist in the middle of a rope with fixed endpoints: approaching the middle, the rope twists, and after the middle, it untwists.

By the twirling formula, $\mu(x)$ is perfectly $G$-equivariant (Fulton & Harris, 1999). The second central moment of the twisted random variable is the covariance: $\text{Cov}_T(Z_x(T)) = \mathbb{E}_T\left[(Z_x(T) - \mu(x))(Z_x(T) - \mu(x))^\top\right]$. The total variance – the trace of the covariance matrix – is a natural measure of equivariance error:

$$\frac{1}{D}\mathbb{E}_{x,T}\left[\|(T^{-1} \circ f \circ T)(x) - \mu(x)\|^2\right] \tag{4}$$

This quantity measures the variance of the twisted predictions around their equivariant mean. An important property is that it is zero if and only if the function is perfectly equivariant. Higher moments of $Z_x(T)$ likewise capture multivariate generalizations of skewness and kurtosis of the equivariance error.

## 2.1 Loss decomposition

Twisting and twirling provide machinery to understand a function's behavior around group actions. We can extend this machinery to analyze losses used to train models under random data augmentation, where each training point is randomly rotated. Let the data distribution $p(x, y)$ and loss function $l : \mathbb{R}^D \times \mathbb{R}^D \to \mathbb{R}$ be invariant to $G$. That is, the joint data distribution $p(x, y)$ for any transformation $T \in G$ satisfies: $p(x, y) = p(T(x), T(y))$ and for any predictions $z$ and targets $y$, and for all $T \in G$: $l(T(z), T(y)) = l(z, y)$. These conditions imply that the loss-optimal model is equivariant, and that: $l((f \circ T)(x), T(y)) = l((T^{-1} \circ f \circ T)(x), y)$. The total loss over all data and transformations is:

$$\mathcal{L}(f) \triangleq \mathbb{E}_{x,y,T}\left[l((T^{-1} \circ f \circ T)(x), y)\right] \tag{5}$$

We perform a Taylor expansion of the total loss around the twirled prediction $\mu(x)$ (averaged over $T$), and obtain terms involving central moments of the twisted random variable:

$$\mathcal{L}(f) = \underbrace{\mathbb{E}_{x,y}[l(\mu(x), y)]}_{\text{twirled prediction error}} + \underbrace{\frac{1}{2}\mathbb{E}_{x,y}\left[\text{tr}\left(\boldsymbol{H}_l(\mu(x), y)\text{Cov}_T[(T^{-1} \circ f \circ T)(x)]\right)\right]}_{\text{equivariance error}} + \mathcal{O}(\|\delta\|^3)$$

where $\delta = (T^{-1} \circ f \circ T)(x) - \mu(x)$, $\boldsymbol{H}_l(\mu, y)$ is the $D \times D$ Hessian matrix of the loss with respect to its first argument, and $\text{Cov}_T$ is a $D \times D$ covariance matrix over the distribution of transformations $T$.

**Proposition 1.** *If $l(z, y) = \frac{1}{D}\|z - y\|^2$ is mean-squared error, then the total loss decomposes as:*

$\mathcal{L}(f) = \mathbb{E}_{x,y}[l(\mu(x), y)] + \frac{1}{D}\mathbb{E}_{x,T}\left[\|(T^{-1} \circ f \circ T)(x) - \mu(x)\|^2\right].$

For MSE loss, our Taylor expansion reduces to a version of bias-variance decomposition. The equivariance error is identical to equation 4 because MSE loss places equal weight on all dimensions. These two terms are central objects of study, so we name them:

$$\mathcal{L}_{\text{mean}} \triangleq \mathbb{E}_{x,y}[l(\mu(x), y)] \tag{6}$$

$$\mathcal{L}_{\text{equiv}} \triangleq \frac{1}{D}\mathbb{E}_{x,T}\left[\|(T^{-1} \circ f \circ T)(x) - \mu(x)\|^2\right] \tag{7}$$

**Percent of loss from equivariance error.** Denoting model parameters as $\theta$, under MSE loss, we can express the total loss exactly as $\mathcal{L}(\theta) = \mathcal{L}_{\text{mean}}(\theta) + \mathcal{L}_{\text{equiv}}(\theta)$. As all three terms are strictly non-negative, this implies:

$$\% \text{ MSE loss from equivariance error} = \frac{\mathcal{L}_{\text{equiv}}(\theta)}{\mathcal{L}(\theta)} \tag{8}$$

**Generalization to convex losses.** We can further define a generalized measure of the percent of loss from equivariance error for any convex loss function with non-negative outputs, such as KL divergence or cross-entropy. By Jensen's inequality, we have $\mathcal{L}_{\text{mean}}(\theta) \leq \mathcal{L}(\theta)$ and both terms are non-negative. Furthermore,

the two terms are equal if and only if the model is exactly equivariant. For convex losses, this motivates defining:

$$\mathcal{L}_{\text{equiv}}(\theta) \triangleq \mathcal{L}(\theta) - \mathcal{L}_{\text{mean}}(\theta) \qquad \text{(for convex losses)} \qquad (9)$$

as the non-negative difference, which is compatible with: % loss from equivariance error $= \mathcal{L}_{\text{equiv}}(\theta)/\mathcal{L}(\theta)$.

**Finite sample estimators.** Denoting twisting as $\hat{Z}_i(x) = (T_i^{-1} \circ f \circ T_i)(x)$, the naive Monte Carlo estimates with $N$ group samples are $\hat{\mu}(x) = \frac{1}{N}\hat{Z}_i(x)$, and for MSE loss $\widehat{\mathcal{L}_{\text{mean}}}(x) = \frac{1}{D}\|\hat{\mu}(x) - y\|^2$, $\widehat{\mathcal{L}_{\text{equiv}}}(x) = \frac{1}{ND}\sum_{i=1}^{N}\|\hat{Z}_i(x) - \hat{\mu}(x)\|^2$. However, $\widehat{\mathcal{L}_{\text{equiv}}}$ is a variance term, so it is biased relative to the true population statistic, unless adjusted by the Bessel correction $N/(N-1)$. In §D.6, we derive unbiased finite-sample estimators for $\mathcal{L}_{\text{mean}}$, $\mathcal{L}_{\text{equiv}}$, and a bias-corrected finite-sample estimator for the percent loss from equivariance error: On 3D molecular tasks, we find that neural networks are typically smooth enough that only five to ten rotation samples are necessary to achieve a stable estimate of the twirled prediction and each loss component (§B.1).

Our derivations provide a principled framework for measuring and understanding degrees of learned equivariance. An important property is that for exactly equivariant architectures, $\mathcal{L}_{\text{equiv}}(\theta) = 0$, so that $\mathcal{L}(\theta) = \mathcal{L}_{\text{mean}}(\theta)$. We remark that sometimes, non-equivariant models may be trained without data augmentation, so that this decomposition may not apply on the training loss. We stress that as long as we aim for these models to behave in an equivariant manner, then the loss decomposition is valid for held-out loss.

## 3 Experiments

To gain insight into the empirical learning behavior of non-equivariant models, we apply our loss decomposition framework to three high-dimensional learning problems on 3D molecules, each with a distinct task and a modern non-equivariant model architecture. For each task, we follow the standard training procedure described in its original publication. The tasks span predictive regression tasks, autoencoding, as well as generative modeling. Notably, all tasks use a mean-squared error loss, so our framework provides an exact decomposition of $\mathcal{L}(f)$ into $\mathcal{L}_{\text{mean}}$ and $\mathcal{L}_{\text{equiv}}$. We report both of these metrics, as well as the percentage of the total loss attributable to the model's lack of equivariance, on a held-out dataset during training. We provide complete details on methods in §D.

- **Neural Interatomic Potential (NNIP)**: We consider force prediction with EScAIP (Qu & Krishnapriyan, 2024), a graph transformer architecture. The model predicts a 3D force vector for each atom based on density functional theory, mapping an input molecule with $N$ atoms to an output in $\mathbb{R}^{3N}$. This task is physically equivariant to the special orthogonal group $SO(3)$ acting on atom coordinates in $\mathbb{R}^3$.

- **Probabilistic Flow Matching**: We study a generative modeling task with Proteína (Geffner et al., 2025), a transformer-based architecture with similarities to AlphaFold3. The model learns to approximate the velocity field of a probability flow that transforms random noise into structured protein backbones. For a molecule with $N$ alpha carbon atoms, the network maps noised atom coordinates and a time $t \in [0,1]$ to a velocity vector in $\mathbb{R}^{3N}$. The learning task is made rotationally equivariant through data augmentation, aligning it with $SO(3)$ acting on atom coordinates in $\mathbb{R}^3$.

- **Denoising Voxelized Atomic Densities**: We analyze a denoising autoencoder task with VoxMol (Pinheiro et al., 2023; Nowara et al., 2025), a non-equivariant 3D convolutional neural network. Molecules are represented as densities in a cubic voxel grid. For a grid length $g$ and $a$ atom types, the input and output are tensors of shape $[g, g, g, a]$. This learning task is made rotationally equivariant through data augmentation using 16 axis-preserving 90-degree rotations of a cube, which do not introduce discretization artifacts due to aliasing. These rotations are a subset of the full octohedral group $O$.

### 3.1 Force field prediction with EScAIP

We trained EScAIP 6M on a subset of SPICE with 950k training examples used by Qu & Krishnapriyan (2024) for 30 epochs with batch size 64. SPICE is a dataset with of small molecule 3D conformers with energies and forces computed by quantum-mechanical density functional theory (Eastman et al., 2024). We varied model size from 1M, 4M and 6M, varied training set size from 950k, 50k, 5k, and 500 (with batch size 1), and varied the optimizer or learning rate. In this task, an equivariant model would output the same yet rotated force prediction, when the input molecule rotates. We observe the following:

- **Equivariance is learned early and quickly, in a manner robust to training set size, model size, and optimizer and learning rate.** The percent validation loss from equivariance error rapidly plummets in the first stage of training to 0.1% within 1k-10k training steps (Fig. 2A-B). Notably, this speed is independent of epoch or training set size - with a 950k training set, this occurs 25% through the first epoch. Training with 500 datapoints with batch size 1, this occurs at the fourth epoch. The dip is least affected by changing model size (Fig. 2E), and most affected by the optimizer and learning rate (Fig. 2F).

- **Equivariance is learned quickly because it is an easier learning task than the main prediction task.** The loss landscape (Fig. 1C) for the equivariance error is much smoother and better conditioned, with a 1,000x lower condition number, than the loss landscape for the twirled prediction error.

- **After a near-universal dip, percent loss from equivariance error can increase mildly.** In the default setting, the percent increases from 0.1% to 0.3%. This is explained by a plateau in the equivariance error while the twirled prediction error continues to decrease (Fig. 2C).

- **Typical models converge to being nearly equivariant, with percent validation loss from equivariance error under 0.1%.** The exception is training on 500 or 5k examples only: equivariance error continues to increase as training progresses, whereas equivariance error decreases in the long-term for larger training set sizes (Fig. 2D, Supp. Fig. 6).

### 3.2 Flow matching with Proteína

The conditional flow matching objective at each time optimizes a mean-squared error loss, which enables us to apply our loss decomposition to study the percent loss from equivariance error. In this task, an equivariant model would output the same yet rotated velocity, when the input noised molecule rotates. Such equivariance is a common desired property for molecular generative models Geffner et al. (2025); Abramson et al. (2024). While many users may care more about final generative quality metrics, the MSE loss plays a critical role in training, monitoring, and in defining the target loss-optimal velocity field. In particular, any non-equivariance in the model's learned generative distribution is caused by a non-equivariant velocity field. This motivates tracking and understanding the percent loss from equivariance error of flow matching models at various $t$, which may enable refining training strategies to improve equivariance.

We trained Proteína at 60M without triangular attention and 400M with triangular attention on the full Protein databank (PDB) dataset with 225k training examples. We also trained models on 1% of the PDB with 2k examples and 0.1% with 200 examples. Flow matching trains a model jointly over $t$, flow matching time, ranging from $t = 0$ for noise and $t = 1$ for data. We measure metrics at $t = 0, 0.2, 0.4, 0.6, 0.8, 0.9, 0.95$, and 0.99, and use red colors for high $t$ close to the data, and blue-purple colors for low $t$ near noise in Figure 3.

We observe that the equivariance learning dip occurs early for all $t$, in a manner robust to training set size and model size. Following the dip at 1k-10k training steps (Fig. 3A), low $t$ (closer to noise) are more equivariant, while high $t$ (closer to data) are less equivariant, which spike following the dip. This holds for the 400M model (Fig. 3E), and 60M model trained on 1% and 0.1% of the PDB (Fig. 3F-H). Interestingly, the dip occurs at the same number of training steps despite different training set sizes, so it occurs 4% through one epoch when trained on the full PDB, but occurs around epoch 53 when trained on 0.1% of the PDB.

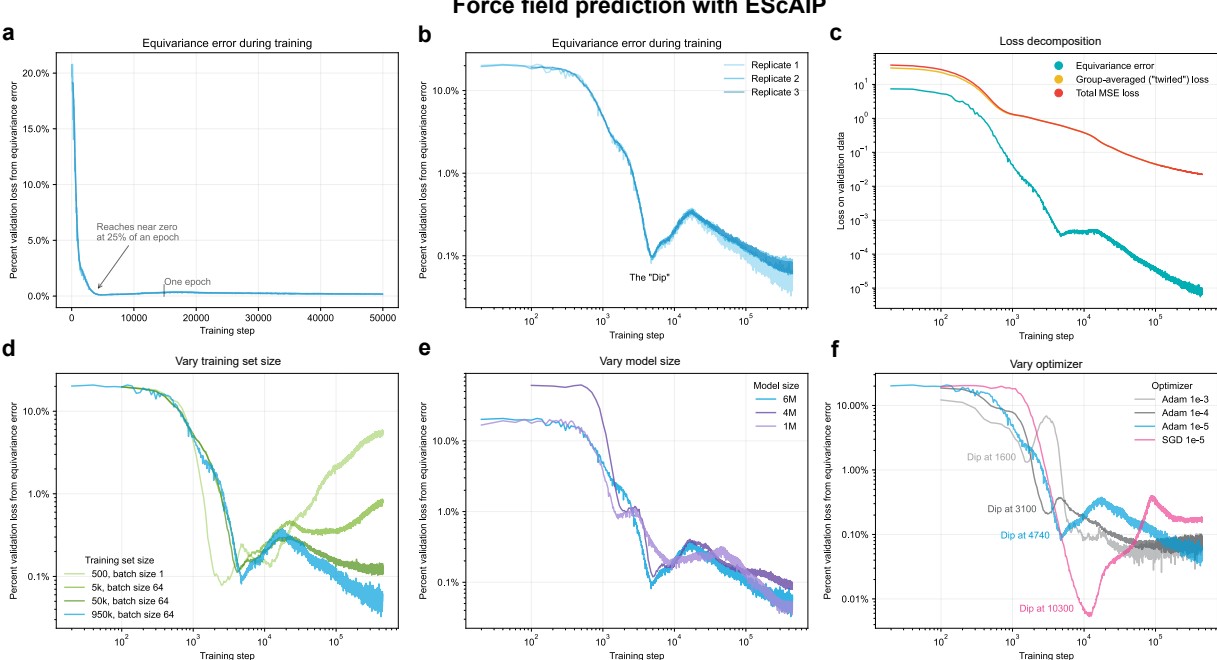

Figure 2: Training dynamics of learning equivariance in EScAIP (Force field prediction). (a-c) Validation losses and percent validation loss from equivariance error during training, early in training (a), with log-log axes (b), and decomposed into separate terms (c). (d-f) Impact of varying training set size (d), model size (e), and optimizer or learning rate (f).

After 1M training steps, the model's equivariance error is low across $t$, with a maximum value around $\sim 6\%$ at $t = 0.90$, though it is lower at the extremes $t = 0.99$ at $3\%$ and $t = 0$ at $0.04\%$ (Fig. 3B). This indicates that the time-conditional velocity field learned by the model is approximately equivariant. Our finding that at 1M training steps, $t = 0.90$ is the least equivariant timestep is relevant for designing training augmentation or test-time strategies for improving equivariance. Notably, this finding highlights the advantage of our loss decomposition framework. Vonessen et al. (2025) measure equivariance error in Proteina by the variance of the normalized twisted prediction, but this is not interpretable as a percent of loss, and thus conflates task difficulty, which gets easier as $t \to 1$, with equivariance error. We correct for this issue, and find that $t = 0.9$ is the most problematic time for non-equivariance, whereas they find $t = 0.5$ instead.

### 3.3 Denoising voxelized atomic densities with VoxMol

We trained VoxMol 111M on GEOM-drugs, a dataset of 3D structures of drug-like molecules with 1.1M training examples. We also trained models on $1\%$ (11k), $10\%$ (110k), $25\%$ (275k), and $50\%$ (550k) examples, and models of varying size: full (111 M parameters), small (28 M), and tiny (7 M). In this autoencoding task, an equivariant model would output a rotated predicted reconstruction when the input molecule rotates; this is a commonly desired property when using the decoder as a generative model Pinheiro et al. (2023). We observe that the equivariance learning dip occurs early for all $t$, in a manner robust to training set and model size. Across the training set sizes, all models rapidly reduce their percent validation loss from equivariance error from an initial $80\%$ to $\sim 2\%$ within 1k-10k training steps (Fig. 4A-B). At 50k training steps, models have around 5-10% validation loss from equivariance error. Beyond 50k training steps, the twirled prediction error continues to decrease while the equivariance error plateaus, or decreases more slowly, below 1e-5 (Fig. 4C).

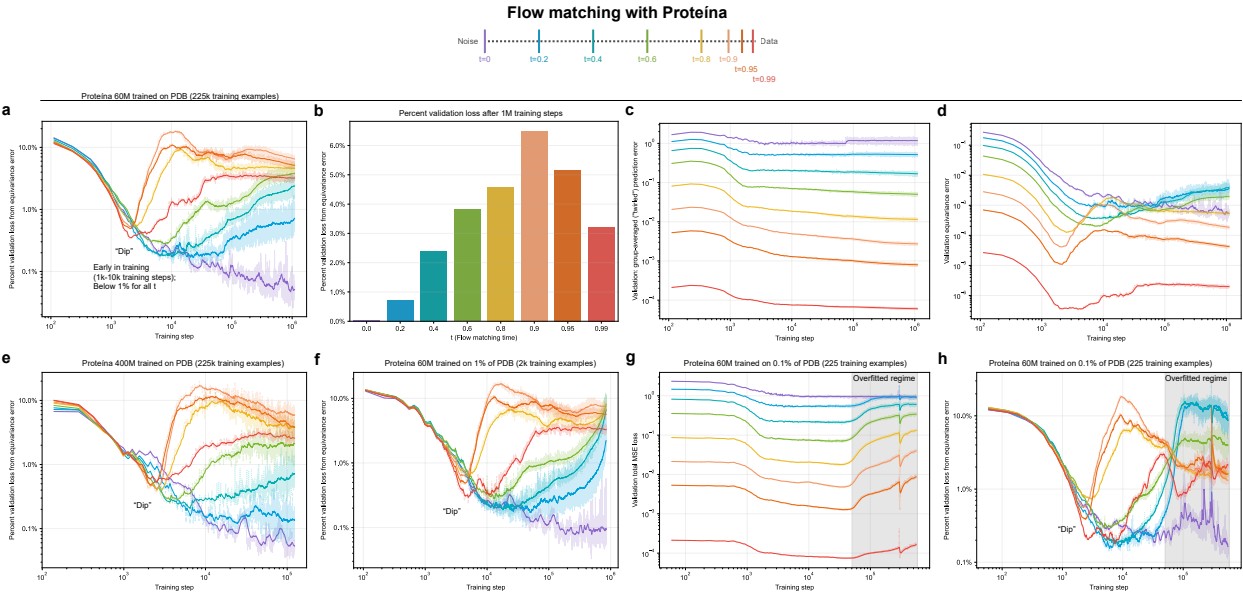

Figure 3: Training dynamics of learning equivariance in Proteína (Flow matching). Colors indicate flow matching time, with noise at $t = 0$ and data at $t = 1$. (a) Percent validation loss from equivariance error during training. (b) Bar plot of the percent validation loss from equivariance error, by flow matching time, at a final checkpoint after 1M training steps. (c-d) Validation losses by training step. (e-h) Impact of varying model size (e), training set size (f-h).

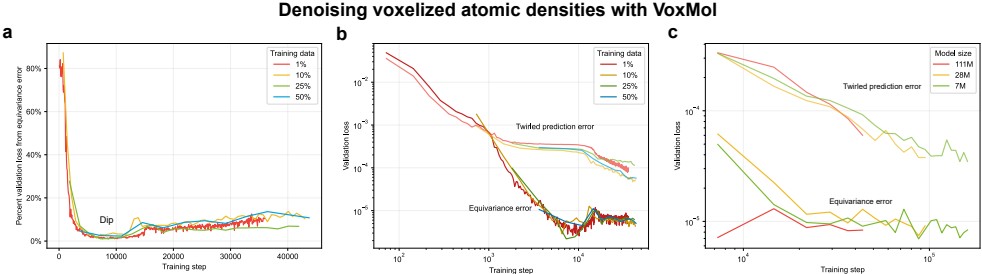

Figure 4: Training dynamics of learning equivariance in VoxMol (Denoising voxelized atomic densities). (a) Percent validation loss from equivariance error during training. (b-c) Validation losses by training step.

## 3.4 Loss Landscape Analysis

To better understand the initial dip, we studied loss landscapes for $\mathcal{L}_{\text{mean}}$ and $\mathcal{L}_{\text{equiv}}$ at early checkpoints (500 steps). We computed the Hessian of each loss on a training batch for a subset of 33k parameters including non-linear layers for EScAIP, 1.5k parameters in Proteína's linear head, and 6.9k parameters in a final layer of VoxMol. For EScAIP, we measured condition numbers around 1e9 for $\mathcal{L}_{\text{mean}}$ and 1e6 for $\mathcal{L}_{\text{equiv}}$ ($\sim$1,000x smaller). For Proteína, we measured 2e10 for $\mathcal{L}_{\text{mean}}$ and 1e8 for $\mathcal{L}_{\text{equiv}}$ ($\sim$100x smaller). For VoxMol, we measured 5e9 and 6e8 respectively ($\sim$10x smaller). We calculate condition numbers for $\mathcal{L}_{\text{mean}}$ and $\mathcal{L}_{\text{equiv}}$ using the largest positive and smallest positive eigenvalues for each loss, and plot them over 20 minibatches in figures 12, 13. For loss landscape plotting, we chose two axes for plotting using the largest positive and smallest positive eigenvector on the total loss, and used the same step size and grid for $\mathcal{L}_{\text{mean}}$ and $\mathcal{L}_{\text{equiv}}$. In both models, we find that $\mathcal{L}_{\text{equiv}}$ has a substantially smoother loss landscape than $\mathcal{L}_{\text{mean}}$ (Fig. 1C).

### 3.5 Do Latent Representations Learn to Respect Equivariance?

The three model architectures studied here have substantial differences (Fig. 1D), yet they display some similarities in their training dynamics of learning equivariance. A natural question is whether their latent representations learn to respect equivariance during training. We provide an analysis here, with further details in §D.

- **EScAIP's latent representation is rotation-invariant.** EScAIP uses rotation-invariant features, and all intermediate layers maintain rotation-invariance. Thus, the final representation is exactly rotation-invariant by design (Fig. 1D). Note that the whole architecture is not invariant or equivariant due to the final prediction head (see eq. 17).

- **Proteína's latent representation is approximately equivariant.** The architecture acts directly on 3D C-alpha coordinates, and does not use rotation-invariant or equivariant features. Its final latent sequence representation is mapped by a linear head into the model output (Fig. 1D). Thus, when the model is empirically approximately equivariant, the final latent is also approximately equivariant.

- **VoxMol's latent representations are not equivariant nor invariant.** Unlike EScAIP and Proteína where first-principles reasoning suffices, we had to study VoxMol empirically. To evaluate if latents were equivariant, for an input molecule and a given rotation, we measured the cosine similarity between the rotated latent, and the latent of the rotated molecule. We found a median of 0.6, comparable to the cosine similarity between latents of different molecules, indicating a lack of equivariance (Fig. 11). To evaluate if latents were invariant, we measured the cosine similarity between the latents of different rotations of the same molecule as 0.64, which is statistically significantly higher but with a small effect size than the cosine similarity between latents of different molecules at 0.58.

## 4 Impact of $\mathcal{L}_{\text{equiv}} < \mathcal{L}_{\text{mean}}$ on Gradients and Parameters

Our empirical results showed that early in training, equivariance error rapidly diminishes, such that $\mathcal{L}_{\text{equiv}}$ is small relative to $\mathcal{L}_{\text{mean}}$. In this section, we investigate empirically and theoretically the connections between equivariance error, gradients and parameters. For non-negative convex losses, the loss is connected to the gradients by an analogous decomposition (eq. 9):

$$\nabla\mathcal{L}(\theta) = \nabla\mathcal{L}_{\text{mean}}(\theta) + \nabla\mathcal{L}_{\text{equiv}}(\theta) \tag{10}$$

$$\nabla\mathcal{L}(\theta) = \nabla\mathcal{L}_{\text{mean}}(\theta) \qquad\qquad \textit{(for exactly equivariant models)} \tag{11}$$

When $\nabla\mathcal{L}_{\text{equiv}}(\theta)$ vanishes, the model is said to follow "equivariant learning dynamics". While gradient norms in general have a complex relationship with loss norms, in §4.1 we show that in certain regions of parameter space, as $\mathcal{L}_{\text{equiv}}$ shrinks, $\nabla\mathcal{L}(\theta)$ can become more similar to $\nabla\mathcal{L}_{\text{mean}}(\theta)$. Experimentally, we find moderate-to-strong correlations between the percent loss from equivariance error, and the similarity of $\nabla\mathcal{L}(\theta) \approx \nabla\mathcal{L}_{\text{mean}}(\theta)$, throughout training.

In §4.2, we consider a parameter decomposition framework from Nordenfors et al. (2025), which shows that when certain assumptions hold, model parameters can be othogonally decomposed:

$$\theta = \theta_{\mathcal{E}} + \theta_{\mathcal{E}\perp} \tag{12}$$

where $\theta_{\mathcal{E}\perp}$ is the model's parameter deviation from the subspace of perfectly equivariant functions $\mathcal{E}$. These assumptions do not always hold, but they do hold for EScAIP's force prediction head, enabling us to prove a quadratic relationship between $\mathcal{L}_{\text{equiv}}$ and parameter deviation from the subspace of exactly equivariant functions for the modern graph transformer. Experimentally, we find the two are closely linked over training with Spearman correlation $= 0.99$. Finally, in general settings where the parameter decomposition does hold, we use our loss decomposition to derive some additional relationships between $\|\theta_{\mathcal{E}\perp}\|$ to $\mathcal{L}_{\text{equiv}}$ for MSE loss.

### 4.1 Equivariance error and gradients

We shall consider the relative loss ratio from equivariance error for non-negative convex losses:

$$\epsilon(\theta) \triangleq \frac{\mathcal{L}_{\text{equiv}}(\theta)}{\mathcal{L}_{\text{mean}}(\theta)} \tag{13}$$

In proposition 2, we consider a mild assumption, commonly satisfied in practice, that the deep neural network is analytic, i.e., it is composed from analytic activation functions, and that the losses are analytic. By standard real analysis results, analyticity implies smoothness on compact subsets of parameter space (such as parameters explored in training; Lemma 7). A function $f$ is $M_f(U)$-smooth in a compact subset $X$ if $\|\nabla f(x) - \nabla f(x')\| \leq M_f(U)\|x - x'\|$ for all $x, x' \in X$. Smoothness in turn can be used to derive a bound on the similarity of $\mathcal{L}(\theta) \approx \nabla\mathcal{L}_{\text{mean}}(\theta)$ in terms of $\epsilon(\theta)$. In practice, the bound can be weak when $M_f(U)$ can be large, and the bound becomes vacuous near saddle points where $\|\nabla\mathcal{L}_{\text{mean}}(\theta)\| \to 0$. Nevertheless, this result sheds light on the structure of the relationship between $\epsilon(\theta)$ and the gradients.

**Proposition 2.** *Let the model $f_\theta$ be analytic (i.e., a deep neural network constructed from analytic activation functions), and suppose the analytic loss satisfies $\mathcal{L}(\theta) = \mathcal{L}_{mean}(\theta) + \mathcal{L}_{equiv}(\theta)$ (i.e., for convex losses via 9). Then, for any compact subset $U$ in the parameter space of $f_\theta$ where $\epsilon(\theta)$ is well-defined (the denominator is non-zero) and that includes an optima $\theta^*$ with $\epsilon(\theta^*) = 0$, there exists a finite constant $M_\epsilon(U) = \sup_{\theta \in U}\|\nabla^2\epsilon(\theta)\|$ such that the approximation $\nabla\mathcal{L}(\theta) \approx \nabla\mathcal{L}_{mean}(\theta)$ holds for all $\theta \in U$ with relative error bounded by:*

$$\frac{\|\nabla\mathcal{L}(\theta) - \nabla\mathcal{L}_{mean}(\theta)\|}{\|\nabla\mathcal{L}_{mean}(\theta)\|} \leq \epsilon(\theta) + \frac{\mathcal{L}_{mean}(\theta)}{\|\nabla\mathcal{L}_{mean}(\theta)\|}\sqrt{2M_\epsilon(U)\epsilon(\theta)} \tag{14}$$

*Proof.* Provided in C.3 □

Near the global optima, we can derive a different bound in proposition 3 using the same assumptions, but employing a theorem known as the Kurdyka-Łojasiewicz (KŁ) inequality (Kurdyka, 1998; Dereich & Kassing, 2021). The Kurdyka-Łojasiewicz (KŁ) inequality is a generalization of the Polyak-Lojasiewicz condition, itself a generalization of convexity, which has been used to study convergence rates of stochastic gradient descent in conditions that are more realistic to deep neural networks (Scaman et al., 2022). The Kurdyka-Łojasiewicz (KŁ) inequality holds locally under mild conditions, only requiring analyticity, and states that there exists a compact local neighborhood $U$ around any critical point $\theta^*$ and constants $c > 0, \alpha \in [1, 2)$ such that, for all $\theta \in U$:

$$\|\nabla f(\theta)\|^2 \geq c|f(\theta) - f(\theta^*)|^\alpha \tag{15}$$

This is mathematically an inequality in the opposite direction as smoothness. Whereas smoothness ensures the function does not change too quickly, the KŁ inequality says the function does not change too slowly, which is important for gradient descent convergence rates. For our purposes, smoothness gives an upper bound on the numerator, while the KŁ inequality provides a lower bound on the denominator. Combining the two gives the ratio bound.

**Proposition 3.** *Let the model $f_\theta$ be analytic (i.e., a deep neural network constructed from analytic activation functions), and suppose $\mathcal{L}(\theta) = \mathcal{L}_{mean}(\theta) + \mathcal{L}_{equiv}(\theta)$ (i.e., for convex losses via 9) is analytic and has a minimum at 0. Then, there exists a compact neighborhood $U$ around the global minimum $\theta^*$ and finite constants $c > 0, M_{\mathcal{L}_{equiv}(\theta)}(U), \alpha \in [1, 2)$ such that for all $\theta \in U$, the approximation $\nabla\mathcal{L}(\theta) \approx \nabla\mathcal{L}_{mean}(\theta)$ holds with relative error bounded by:*

$$\frac{\|\nabla\mathcal{L}(\theta) - \nabla\mathcal{L}_{mean}(\theta)\|}{\|\nabla\mathcal{L}_{mean}(\theta)\|} \leq \sqrt{\frac{2M_{\mathcal{L}_{equiv}(\theta)}(U)}{c}} \cdot \sqrt{\frac{\mathcal{L}_{equiv}(\theta)}{\mathcal{L}_{mean}(\theta)^\alpha}} \tag{16}$$

*Proof.* Provided in §C.4. □

**Experimental investigation.** Our propositions 2, 3 show that in certain conditions, a function of the loss ratio upper bounds the gradient norm ratio. We empirically investigated this by measuring the loss ratio and the gradient norm ratio during training, which we plot in §B.3. We find strong log-log correlations between the loss ratio and the gradient norm ratio of Pearson R = 0.75 over training in EScAIP, and statistically significant R = 0.23 to 0.95 for Proteína across all times. Interestingly, we observe stronger correlations at smaller "lag" times, suggestive of stable "coefficients" within basins, with a sparse number of transitory windows where the models seem to "move between basins". Collectively, these experimental results provide complementary insight into the relationship between relative equivariance error and model gradients.

### 4.2 Equivariance error and deviation from equivariant parameter subspace

In the proceeding analysis, we adopt Nordenfors et al. (2025)'s mathematical framework for analyzing neural network parameters in terms of equivariant and non-equivariant parameter subspaces; under certain conditions, model parameters can be decomposed into orthogonal components: $\theta = \theta_{\mathcal{E}} + \theta_{\mathcal{E}\perp}$. This framework relies on three assumptions: (i) the symmetry group is compact and acts on finite-dimensional hidden spaces; (ii) the neural net non-linearities are equivariant; and (iii) the loss is invariant. Under these conditions, the total parameter space is an inner product space, and the subspace of perfectly equivariant functions $\mathcal{E}$ is a linear subspace, which together enable the orthogonal decomposition $\theta = \theta_{\mathcal{E}} + \theta_{\mathcal{E}\perp}$. This framework can be applied to a broad class of modern neural network operations and architectures, including fully connected layers with non-linearities, convolutions, residual connections, and attention layers. It also includes a broad class of symmetry groups including $SO(3)$ and all groups studied in this work. We provide more detail in §A.1 and refer the interested reader to Nordenfors et al. (2025).

Our scope and contributions here are as follows. Assumptions (i-iii) and the parameter decomposition $\theta = \theta_{\mathcal{E}} + \theta_{\mathcal{E}\perp}$ do not always hold, but they do hold for EScAIP's force prediction head, enabling us to prove a quadratic relationship between $\mathcal{L}_{\text{equiv}}$ and parameter deviation from the subspace of exactly equivariant functions for the modern graph transformer. Furthermore, in general settings where the parameter decomposition does hold, Nordenfors et al. (2025) studies its implications, such as models following approximately equivariant learning dynamics when $\|\theta_{\mathcal{E}\perp}\|$ is small. In this general setting, we apply our loss decomposition to derive some relationships between $\|\theta_{\mathcal{E}\perp}\|$ to $\mathcal{L}_{\text{equiv}}$.

What is the relationship between $\mathcal{L}_{\text{equiv}}$ and $\|\theta_{\mathcal{E}\perp}\|$ when $\theta = \theta_{\mathcal{E}} + \theta_{\mathcal{E}\perp}$? In general for neural networks, $\mathcal{L}_{\text{equiv}}$ is a complex, highly non-linear function of $\theta$. However, we know that $\mathcal{L}_{\text{equiv}}$ is non-negative, continuous, and equal to zero iff $\theta_{\mathcal{E}\perp} = 0$. By these properties, we know that if $\|\theta_{\mathcal{E}\perp}\|$ is small, then $\mathcal{L}_{\text{equiv}}$ is small. More formally, for any $\epsilon > 0$, there exists a $\delta > 0$ such that if the parameter deviation is small ($\|\theta_{\mathcal{E}\perp}\| < \delta$), then the equivariance error is also small ($\mathcal{L}_{\text{equiv}} < \epsilon$).

The EScAIP architecture is a modern graph transformer architecture that achieved strong results on NNIP energy and force prediction tasks, and satisfies assumptions (i-iii). The EScAIP architecture uses rotation-invariant features derived from an input molecular graph. Its hidden representations for atoms and edges, denoted $\boldsymbol{h}$, are rotation-invariant throughout the network. Force prediction outputs a 3D force vector at each atom in a molecule. For a single atom with a set of 3D edge vectors $E$ (the vectors pointing from one atom to another atom) in a molecule $\boldsymbol{x}$, EScAIP predicts force vectors as:

$$\begin{bmatrix} o_x \\ o_y \\ o_z \end{bmatrix} = \sum_{e \in E} \begin{bmatrix} e_x \cdot \mathbf{w_x}^\intercal \mathbf{h}(\mathbf{e}, \mathbf{x}) \\ e_y \cdot \mathbf{w_y}^\intercal \mathbf{h}(\mathbf{e}, \mathbf{x}) \\ e_z \cdot \mathbf{w_z}^\intercal \mathbf{h}(\mathbf{e}, \mathbf{x}) \end{bmatrix} \tag{17}$$

where $\boldsymbol{e} \in \mathbb{R}^3$ is a 3D edge vector, $\mathbf{h}(\mathbf{e}, \mathbf{x}) \in \mathbb{R}^h$ is the last hidden representation of the edge $\boldsymbol{e}$ in molecule $\boldsymbol{x}$, and $W = [\mathbf{w_x}, \mathbf{w_y}, \mathbf{w_z}]$, where each $\mathbf{w} \in \mathbb{R}^h$, are the parameters for a linear head with no bias. The 3D edge vectors $\boldsymbol{e}$ are rotation-equivariant with respect to the input molecule, while the hidden representation $\mathbf{h}(\mathbf{e})$ is rotation-invariant to the input molecule, but composing these to form the output prediction generally breaks both invariance and equivariance.

In particular, force predictions are equivariant if and only if the scalar projections of the hidden features are independent of the coordinate axis, i.e., $\mathbf{w_x}^\intercal \mathbf{h}(\mathbf{e}, \mathbf{x}) = \mathbf{w_y}^\intercal \mathbf{h}(\mathbf{e}, \mathbf{x}) = \mathbf{w_z}^\intercal \mathbf{h}(\mathbf{e}, \mathbf{x})$, for all inputs. Under the mild assumption of a non-degenerate learned embedding function $\mathbf{h}(\mathbf{e}, \mathbf{x})$, such that the set of all possible hidden vectors spans the feature space, this condition holds if and only if the parameter vectors themselves are identical: $\mathbf{w_x} = \mathbf{w_y} = \mathbf{w_z}$. This condition defines the subspace $\mathcal{E}$ for the EScaIP architecture. Using this, we decompose $\boldsymbol{W} = \mathbf{W}_\mathcal{E} + \mathbf{W}_{\mathcal{E}\perp}$ with an equivariant part $\mathbf{W}_\mathcal{E} = [\bar{\boldsymbol{w}}, \bar{\boldsymbol{w}}, \bar{\boldsymbol{w}}] \in \mathcal{E}$ where $\bar{\boldsymbol{w}} = \frac{1}{3}(\mathbf{w_x} + \mathbf{w_y} + \mathbf{w_z})$, and a non-equivariant part $\mathbf{W}_{\mathcal{E}\perp} = [\boldsymbol{d}_x, \boldsymbol{d}_y, \boldsymbol{d}_z] \in \mathcal{E}\perp$ where $\boldsymbol{d}_x = \mathbf{w_x} - \bar{\boldsymbol{w}}$, and same for $y, z$.

With this setup, we can now establish that the equivariance error of the EScaIP architecture has a quadratic relationship with the magnitude of the parameter deviation from $\mathcal{E}$, the space of perfectly equivariant functions.

**Theorem 4.** *For the EScAIP architecture trained with mean-squared error loss on a non-degenerate dataset, for any fixed set of upstream parameters $\theta \setminus \boldsymbol{W}$, there exist positive constants $0 < \lambda_{min} \le \lambda_{max}$ (which depend on the model architecture, data distribution, and other parameters $\theta \setminus \boldsymbol{W}$) such that:*

$$\lambda_{min} \cdot \|\mathbf{W}_{\mathcal{E}\perp}\|_F^2 \le \mathcal{L}_{equiv}(\theta) \le \lambda_{max} \cdot \|\mathbf{W}_{\mathcal{E}\perp}\|_F^2 \tag{18}$$

*Proof.* Provided in C.5. □

**Experimental investigation.** In §B.4, we empirically plot the force prediction head's deviation from the mean, vs. equivariance error, and find a Pearson correlation of 0.94 and Spearman correlation of 0.99 over training, which supports our conclusion.

The preceding analysis can be generalized to a broader class of neural networks. Applying a Taylor expansion to $\mathcal{L}_{\text{equiv}}(\theta)$ for the neural net $f$ on an input $x$, we have: $f(x; \theta_\mathcal{E} + \theta_{\mathcal{E}\perp}) = f(x; \theta_\mathcal{E}) + J_{\theta_{\mathcal{E}\perp}} f(x; \theta_\mathcal{E}) \cdot \theta_{\mathcal{E}\perp} + \mathcal{O}(\|\theta_{\mathcal{E}\perp}\|^2)$ where $J_{\theta_{\mathcal{E}\perp}} f(x; \theta_\mathcal{E})$ is the Jacobian of the network output with respect to parameter components $\theta_{\mathcal{E}\perp}$, evaluated at $\theta_\mathcal{E}$. The key structure, analogous to the EScaIP argument, is the decomposition of the neural net output into a purely equivariant term, and a term linear in $\theta_{\mathcal{E}\perp}$, as well as a remainder term in this setting. With this setup, for a broad class of neural network architectures, we can relate locally near $\mathcal{E}$ that $\mathcal{L}_{\text{equiv}}$ is quadratic in $\|\theta_{\mathcal{E}\perp}\|$ (Thm. 5), and its grad norm is linear in $\|\theta_{\mathcal{E}\perp}\|$ (Thm. C.7).

**Theorem 5.** *For any neural network whose parameters can be expressed as $\theta = \theta_\mathcal{E} + \theta_{\mathcal{E}\perp}$ with $\theta_\mathcal{E} \in \mathcal{E}$ and $\theta_{\mathcal{E}\perp} \in \mathcal{E}\perp$, and for equivariance error $\mathcal{L}_{equiv}$ defined by the variance of the output with respect to transformations, there exist positive constants $0 < \lambda_{min} \le \lambda_{max}$ such that for a non-degenerate dataset, using $\|\cdot\|$ to denote $L_2$-norm:*

$$\lambda_{min}\|\theta_{\mathcal{E}\perp}\|^2 + \mathcal{O}(\|\theta_{\mathcal{E}\perp}\|^3) \le \mathcal{L}_{equiv}(\theta) \le \lambda_{max}\|\theta_{\mathcal{E}\perp}\|^2 + \mathcal{O}(\|\theta_{\mathcal{E}\perp}\|^3) \tag{19}$$

*Proof.* Provided in C.6. □

**Theorem 6.** *Under the same conditions as Thm. 5, the norm of the gradient of the equivariance loss with respect to the non-equivariant parameters is bounded by the deviation itself. Specifically, there exists a constant $C$ such that:*

$$\|\nabla_{\theta_{\mathcal{E}\perp}} \mathcal{L}_{equiv}(\theta)\| \le C \cdot \|\theta_{\mathcal{E}\perp}\|$$

*Proof.* Provided in C.7. □

## 5 Related Work

Prior work have measured learned equivariance with a wide variety of approaches (Kvinge et al., 2022; Karras et al., 2021; Geffner et al., 2025; Qu & Krishnapriyan, 2024; Gruver et al., 2023; Nowara et al., 2025; Fuchs et al., 2020), but to our knowledge, this work is the first to derive a measure of equivariance error that is interpreted as a percent of loss. Notably, many prior measures effectively estimate equivariance error as a pairwise deviation using only two samples per datapoint, whereas we estimate variance around a mean

using enough samples of the twisted prediction as necessary to obtain stable estimates. Vonessen et al. (2025) use the variance of the normalized twisted prediction, but this is not interpretable as a percent of loss. They study flow matching, but their metric conflates task difficulty, which gets easier as $t \to 1$, with equivariance error. We correct for this issue, and find that $t = 0.9$ is the most problematic time for non-equivariance, whereas they find $t = 0.5$ instead. Canez et al. (2024) find that relaxing architectures from exact equivariance improves loss landscape conditioning and achieves better loss than perfectly equivariant architectures on image super-resolution and fluid dynamics modeling.

Twirling serves as a simple yet powerful postprocessing operation to transform any learned function into an equivariant one at test time. Ideas like this have been explored in Pozdnyakov & Ceriotti (2023); Nordenfors et al. (2025). Preprocessing inputs to a canonical frame is another simple yet powerful postprocessing operation to convert any function into an equivariant one (Mondal et al., 2023; Gandikota et al., 2021).

## 6 Discussion

In this work, we found that 3D-rotational equivariance is learned easily and quickly. We described a two-phase learning dynamic: initially, models rapidly learn equivariance. This occurs because learning equivariance is an easier task, with a smoother and better-conditioned loss landscape, than the main prediction task. After training, the final percent loss from equivariance error is small for all models, but it is notably smaller for EScAIP at 0.006% than for Proteína and VoxMol ($< 5\%$). While all of these loss penalties are small, and easily remedied by test-time postprocessing techniques like twirling or input frame canonicalization, this observation may also motivate research on architecture design to narrow this gap.

Intriguingly, equivariance is learned rapidly despite significant differences in model architectures. EScAIP is "nearly equivariant", as it becomes exactly equivariant with only a small change to its final linear head, yet its initial dip occurs just as quickly as Proteína and VoxMol, which are distant from being architecturally equivariant. It is also interesting that each model's latents learn (or fail to learn) to respect symmetries in different ways.

Our work establishes a principled and unified framework for quantifying equivariance error for non-negative convex losses. We focused our empirical study on 3D rotations, as this is a physically important symmetry group for biomolecules, but other symmetry groups may be easier or harder to learn. Looking forward, our framework could be used to study the learning dynamics of equivariance on other symmetry groups.

## Acknowledgements

We thank Pan Kessel and Saeed Saremi for helpful discussions.

## Code Availability

We provide code at https://github.com/genentech/equivariance_learning. Our code simply adds callbacks to compute equivariance metrics during training on top of the original EScAIP, Proteína, and VoxMol codebases.

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

# A    Appendix

## A.1    Parameter space decomposition

Here, we describe in greater detail Nordenfors et al. (2025)'s mathematical framework for analyzing the geometry of neural network parameters in terms of equivariant and non-equivariant parameter subspaces. This framework relies on three assumptions: (i) the symmetry group is compact and acts on finite-dimensional hidden spaces; (ii) the neural net non-linearities are equivariant; and (iii) the loss is invariant.

In practice, conditions (i) and (iii) are satisfied by many settings. Notably, the group $SO(3)$ of 3D rotations is compact, but $SE(3)$ which includes translations is not compact; but this is readily handled by restricting modeling to positionally centered data. Finite-dimensional hidden spaces is easily satisfied by neural networks. Belief that a task is equivariant or invariant implies that the correct loss to use must be invariant. Condition (ii) is more setting-specific; this is satisfied in the VoxMol setting with voxel atomic densities with rotations, and non-linearities that act pixel-wise, but this is not generally satisfied by element-wise non-linearities on linear transformations of 3D coordinates.

In the scope of this manuscript, we use Nordenfors et al. (2025)'s framework to study the linear head of the EScAIP architecture, which does satisfy the conditions. We also use the framework to extend Proposition 3.14 in Nordenfors et al. (2025), which describes how models follow approximately equivariant learning dynamics when $\theta_{\mathcal{E}\perp}$ is small. In our propositions 5, 6, we extend this with our loss decomposition framework, and derive a relationship between $\theta_{\mathcal{E}\perp}$ and $\mathcal{L}_{\mathrm{equiv}}$.

The foundation of this framework is the representation of a network's parameters in all of its linear layers as a point in a high-dimensional vector space, denoted $\mathcal{H}$. This captures the dominant set of learnable parameters when non-linearities are fixed. The space is formally constructed as the direct sum of the parameter spaces for each individual layer: $\mathcal{H} = \bigoplus_i \mathrm{Hom}(X_i, X_{i+1})$. Specific network architectures are assumed to have parameters in an affine subspace $\mathcal{L} \subseteq \mathcal{H}$, referred to as the space of "admissible layers". This setup is shown by construction to be expressive and capable of describing many modern neural network architectures and operations, including fully connected layers, convolutions, residual connections, and attention layers.

To define equivariance for a multi-layer network, the framework supposes that the symmetry group $G$ acts on all input, hidden, and output spaces $(X_0, X_1, ..., X_L)$ through a series of representations, $\rho_i$. With this setup, the set of all parameter configurations where each linear layer is individually equivariant forms a linear subspace of $\mathcal{H}$, denoted $\mathcal{H}_G$. This set is a linear subspace because the group actions $\rho_i(g)$ is a linear operator, which means any linear combination of equivariant linear maps remains equivariant. For instance in the setting of rotations on 3D molecules, consider a linear layer with matrix $A$ with a rotation matrix $R$ – if it is equivariant, we have $ARx = RAx$. If $A$ and $B$ are both equivariant to $R$, then $C = c_1 A + c_2 B$ is also equivariant to $R$: $RCx = R(c_1 A + c_2 B)x = (c_1 A + c_2 B)Rx = CRx$. $\mathcal{H}_G$ is thus a linear subspace that is closed under addition and scalar multiplication.

Algebraic manipulations show that $TC_i x = C_i Tx$, using:

$$\begin{aligned}
TC_i x &= T(c_1 A_i + c_2 B_i)x \\
&= c_1 A_i Tx + c_2 B_i Tx \\
&= (c_1 A_i + c_2 B_i)Tx \\
&= C_i Tx
\end{aligned}$$

This subspace's linearity follows from the group's actions being linear transformations.

The parameters that are both architecturally admissible and perfectly equivariant then lie in the intersection of these spaces, $\mathcal{E} = \mathcal{L} \cap \mathcal{H}_G$. It further follows that if non-linearities are equivariant (condition ii), then the entire neural network function is equivariant when its parameters are in $\mathcal{E}$.

This geometric structure guarantees that any admissible parameters $\theta$ in $\mathcal{L}$ can be uniquely decomposed via orthogonal projection into two components: $\theta = \theta_{\mathcal{E}} + \theta_{\mathcal{E}\perp}$. This is possible because $\mathcal{H}$ being an inner product space allows for a unique projection onto the tangent space of the subspace $\mathcal{E}$. The component $\theta_{\mathcal{E}}$ is the

projection of the parameters onto the subspace of equivariant functions ($\mathcal{E}$), while $\theta_{\mathcal{E}_\perp}$ is the component in the orthogonal complement of this subspace, representing deviation from perfect equivariance.

# B    Supplementary Analyses & Figures

## B.1    Sensitivity of percent loss from equivariance error to number of rotations used for estimation

Here, we plot bootstrapped estimates of standard error of percent loss from equivariance error, for a model with 4 percent loss from equivariance error, with varying number of rotations used to estimate the statistic. The model is a 60M Proteina model early in training at 500 steps, evaluated at $t = 0.5$. We compute loss metrics across 100 randomly sampled 3D rotations, and use this metrics set to derive subsampled bootstrap statistics for lower number of rotations. We find that the Proteina model is smooth enough, even early in training, over the group of 3D rotations. Our practical choice of estimating percent loss from equivariance error with 10 rotations has low standard error around 0.0015 compared to the measured percent loss from equivariance error of 0.04.

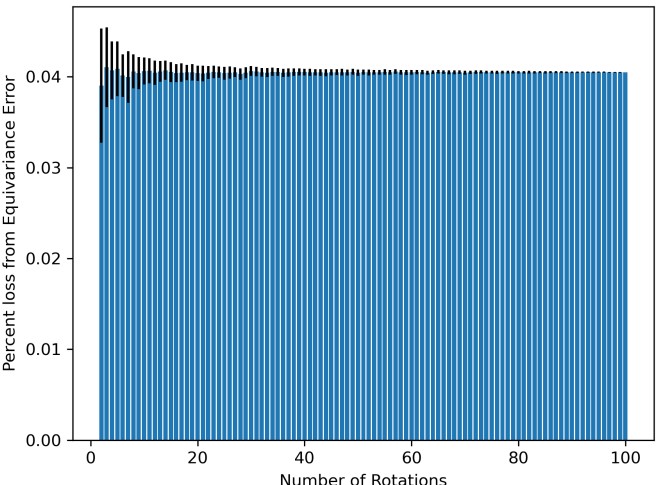

Figure 5: Bootstrapped estimate of standard error of percent loss from equivariance error, for a model with 4 percent loss from equivariance error.

## B.2    Validation loss curves for EScAIP, by training set size

Here, we plot validation loss curves for EScAIP, split into total MSE loss, group-averaged loss, and equivariance error, for models trained on different training set sizes. These results highlight that across a variety of training set sizes, group-averaged loss dominates the total loss. At smaller training set sizes, equivariance error rises towards the end of training, suggesting some type of overfitting effect, while at larger training set sizes, equivariance error continues to shrink.

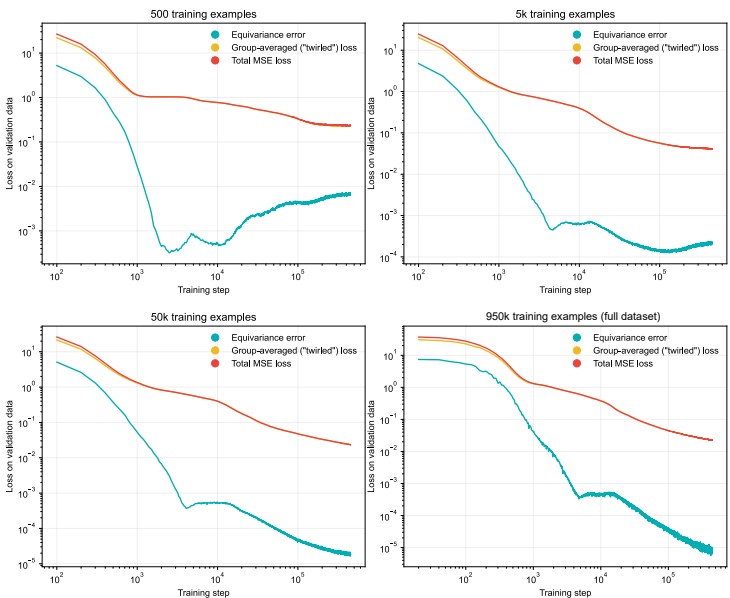

Figure 6: EScAIP: Validation loss curves over training, varied by training set size.

## B.3 Gradient norm ratios vs. loss ratios

Here, we plot the loss ratio (percent loss from equivariance error) to the gradient norm ratio: the ratio of the gradient norm of the equivariance error, to the gradient norm of the group-averaged loss, over training. In general, gradient norms have a complex relationship with loss norms. In propositions 2 and 3, we show that under certain local smoothness assumptions, the gradient norm ratio can be controlled by the loss ratio; however, this smoothness assumption may not apply in practice. With these empirical plots, we find a generally strong correlation between loss ratio and gradient norm ratio.

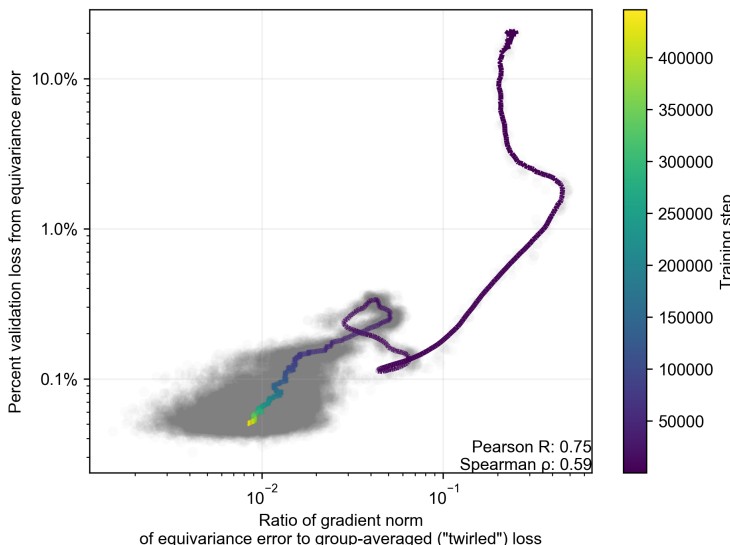

Figure 7: EScAIP: Percent validation loss from equivariance error vs. grad norm ratio, over training. Colored line indicates smoothed exponential moving average, colored by training step.

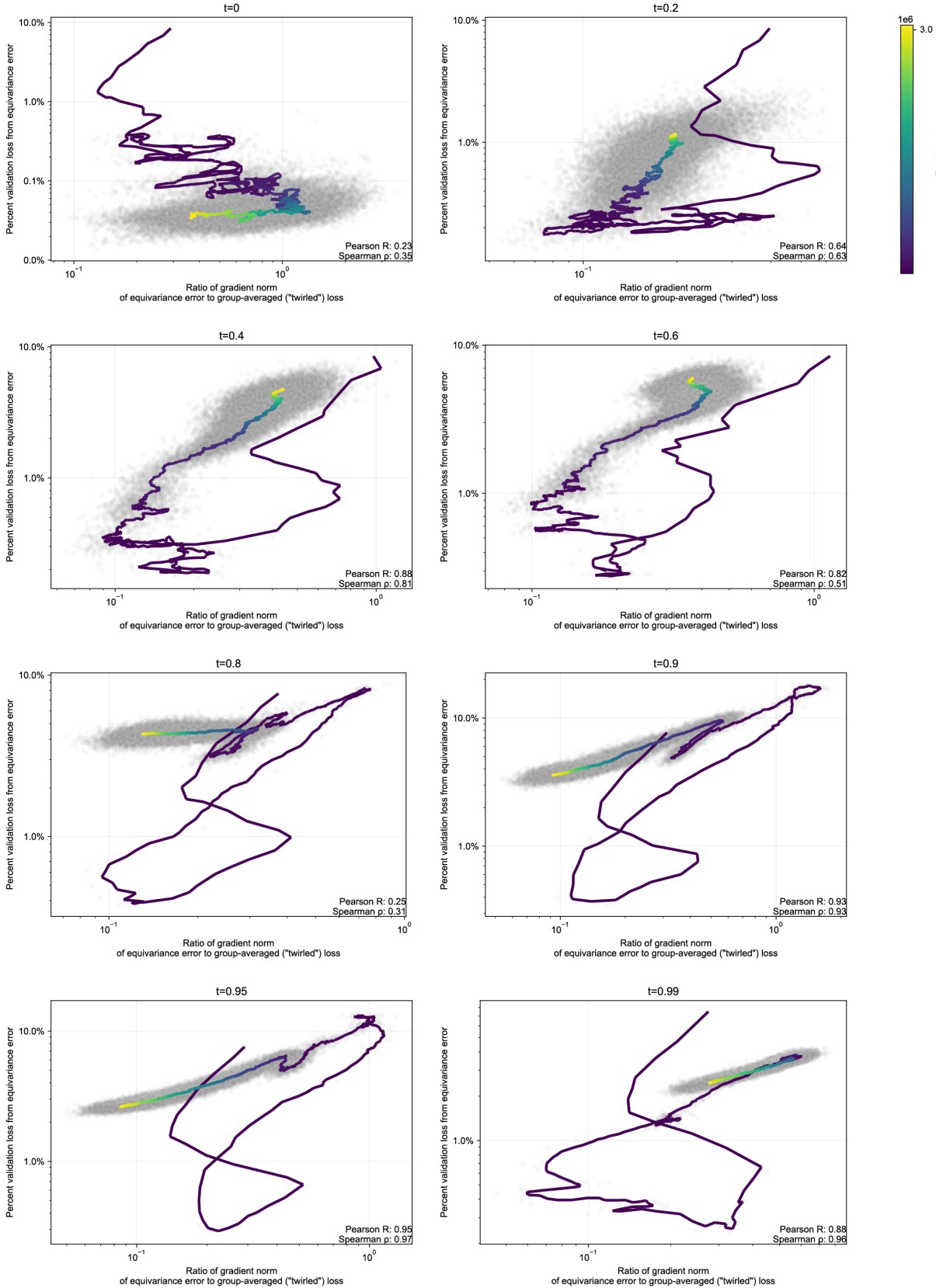

Figure 8: Proteína: Percent validation loss from equivariance error vs. grad norm ratio, over training, by flow matching time. Colored line indicates smoothed exponential moving average, colored by training step.

### B.4 Variance of EScAIP's linear force head controls equivariance error

Here, we provide plots of the variance, or deviation from the mean, of EScAIP's linear force head, over training time, and compared to percent loss from equivariance error. Our proposition 4 states that equivariance error is controlled quadratically by the linear head's deviation from its mean. Empirically, we find strong agreement between the two metrics, with Pearson correlation = 0.94, and Spearman correlation = 0.99 over training.

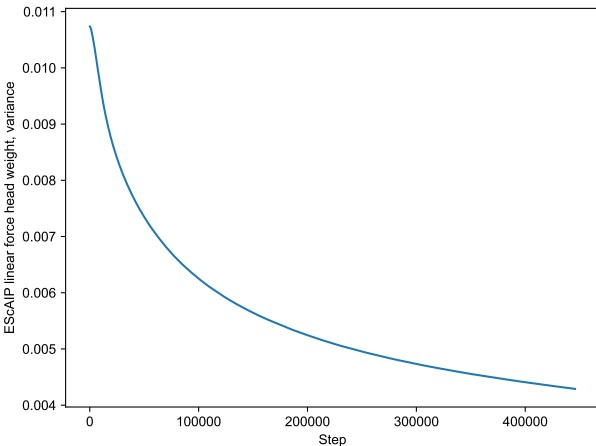

Figure 9: EScAIP: Variance of linear force head weights, by training time.

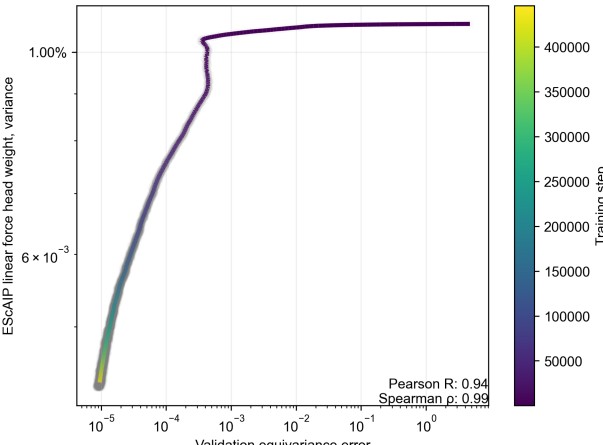

Figure 10: EScAIP: Variance of linear force head weights, vs. percent validation loss from equivariance error, over training.

## C Proofs

### C.1 Proof of Proposition 1

**Proposition.** *If $l(z, y) = \frac{1}{D}\|z - y\|^2$ is mean-squared error, then the total loss decomposes as:*

$$\mathcal{L}(f) = \underbrace{\mathbb{E}_{x,y}[l(\mu(x), y)]}_{prediction\ error} + \frac{1}{D}\mathbb{E}_{x,y}\left[\underbrace{\sum_{i=1}^{D} Var_T[(T^{-1} \circ f \circ T)(x)_i]}_{equivariance\ error}\right] \tag{20}$$

*Proof.* For mean-squared error, the Hessian is constant: $H_l(z, y) = \frac{2}{D}I$ where $I$ is the $D \times D$ identity matrix. Furthermore, higher-order derivatives are zero, so the decomposition has no additional terms. The equivariance error simplifies as:

$$\frac{1}{2}\mathbb{E}_{x,y}\left[\text{tr}\left(\left(\frac{2}{D}I\right)\text{Cov}_T[\dots]\right)\right] = \frac{1}{D}\mathbb{E}_{x,y}\left[\text{tr}\left(\text{Cov}_T[\dots]\right)\right] \tag{21}$$

$\square$

### C.2 Lemma: Analytic functions are smooth on compact sets

**Lemma 7.** *Let $f : \mathcal{X} \to \mathbb{R}$ be a real-analytic function. Then, on any compact subset $X \subset \mathcal{X}$, there exists a finite constant $M > 0$ such that $f$ is $M$-smooth on $X$; that is,*

$$\frac{1}{2}\|\nabla f(x)\|^2 \le M|f(x) - f(x^*)| \tag{22}$$

*for some minimum $x^* \in X$.*

*Proof.* Since $f$ is analytic, all derivatives of $f$ exist and are continuous. In particular, its Hessian $\nabla^2 f(x)$ is continuous on $\mathcal{X}$. By the extreme value theorem, any continuous function on a compact set attains a finite supremum. Therefore,

$$M \triangleq \sup_{x \in X} \|\nabla^2 f(x)\|_2 < \infty. \tag{23}$$

This boundedness of the Hessian implies that $\nabla f$ is Lipschitz continuous with constant $M$ on $X$, which is the $M$-smoothness condition. The inequality $\frac{1}{2}\|\nabla f(x)\|^2 \le M\|f(x) - f(x')\|$ follows from integrating the gradient-Lipschitz property along the line segment between $x$ and $x^*$, or by standard smoothness results. $\square$

### C.3 Proof of Proposition 2

**Proposition.** *Let the model $f_\theta$ be analytic (i.e., a deep neural network constructed from analytic activation functions), and suppose the analytic loss satisfies $\mathcal{L}(\theta) = \mathcal{L}_{mean}(\theta) + \mathcal{L}_{equiv}(\theta)$ (i.e., for convex losses via 9). Then, for any compact subset $U$ in the parameter space of $f_\theta$ where $\epsilon(\theta)$ is well-defined (the denominator is non-zero) and that includes an optima $\theta^*$ with $\epsilon(\theta^*) = 0$, there exists a finite constant $M_\epsilon(U) = \sup_{\theta \in U} \|\nabla^2 \epsilon(\theta)\|$ such that the approximation $\nabla \mathcal{L}(\theta) \approx \nabla \mathcal{L}_{mean}(\theta)$ holds for all $\theta \in U$ with relative error bounded by:*

$$\frac{\|\nabla\mathcal{L}(\theta) - \nabla\mathcal{L}_{mean}(\theta)\|}{\|\nabla\mathcal{L}_{mean}(\theta)\|} \le \epsilon(\theta) + \frac{\mathcal{L}_{mean}(\theta)}{\|\nabla\mathcal{L}_{mean}(\theta)\|}\sqrt{2M_\epsilon(U)\epsilon(\theta)} \tag{24}$$

*Proof.* Using $\mathcal{L}(\theta) = \mathcal{L}_{\text{mean}}(\theta) + \mathcal{L}_{\text{equiv}}(\theta)$, the total loss is $\mathcal{L}(\theta) = (1 + \epsilon(\theta))\mathcal{L}_{\text{mean}}(\theta)$. Differentiate with the product rule:

$$\nabla\mathcal{L}(\theta) = \nabla[(1 + \epsilon(\theta))\mathcal{L}_{\text{mean}}(\theta)] \tag{25}$$

$$= \nabla\epsilon(\theta)\mathcal{L}_{\text{mean}}(\theta) + (1 + \epsilon(\theta))\nabla\mathcal{L}_{\text{mean}}(\theta) \tag{26}$$

$$\nabla\mathcal{L}(\theta) - \nabla\mathcal{L}_{\text{mean}}(\theta) = \epsilon(\theta)\nabla\mathcal{L}_{\text{mean}}(\theta) + \mathcal{L}_{\text{mean}}(\theta)\nabla\epsilon(\theta) \tag{27}$$

Now, we bound the norm of this difference using the triangle inequality:

$$\|\nabla\mathcal{L}(\theta) - \nabla\mathcal{L}_{\text{mean}}(\theta)\| \le \epsilon(\theta)\|\nabla\mathcal{L}_{\text{mean}}(\theta)\| + \mathcal{L}_{\text{mean}}(\theta)\|\nabla\epsilon(\theta)\| \tag{28}$$

The model and losses are analytic, so by lemma 7, there exists a finite $M_\epsilon(U)$ such that $\epsilon(\theta)$ is $M_\epsilon(U)$-smooth over the compact parameter region $U$. Specifically, $M_\epsilon(U) = \sup_{\theta \in U} \|\nabla^2 \epsilon(\theta)\|$. Applying the smoothness property $\|\nabla \epsilon(\theta)\| \leq \sqrt{2M_\epsilon(U)\epsilon(\theta)}$ relative to the optima $\theta^*$ with $\epsilon(\theta^*) = 0$, we obtain the final result:

$$\frac{\|\nabla \mathcal{L}(\theta) - \nabla \mathcal{L}_{\mathrm{mean}}(\theta)\|}{\|\nabla \mathcal{L}_{\mathrm{mean}}(\theta)\|} \leq \epsilon(\theta) + \frac{\mathcal{L}_{\mathrm{mean}}(\theta)}{\|\nabla \mathcal{L}_{\mathrm{mean}}(\theta)\|} \sqrt{2M_\epsilon(U)\epsilon(\theta)} \tag{29}$$

$\square$

### C.4   Proof of Proposition 3

**Proposition.** *Let the model $f_\theta$ be analytic (i.e., a deep neural network constructed from analytic activation functions), and suppose $\mathcal{L}(\theta) = \mathcal{L}_{mean}(\theta) + \mathcal{L}_{equiv}(\theta)$ (i.e., for convex losses via 9) is analytic and has a minimum at 0. Then, there exists a compact neighborhood $U$ around the global minimum $\theta^*$ and finite constants $c > 0, M_{\mathcal{L}_{equiv}(\theta)}(U), \alpha \in [1, 2)$ such that for all $\theta \in U$, the approximation $\nabla \mathcal{L}(\theta) \approx \nabla \mathcal{L}_{mean}(\theta)$ holds with relative error bounded by:*

$$\frac{\|\nabla \mathcal{L}(\theta) - \nabla \mathcal{L}_{mean}(\theta)\|}{\|\nabla \mathcal{L}_{mean}(\theta)\|} \leq \sqrt{\frac{2M_{\mathcal{L}_{equiv}(\theta)}(U)}{c}} \cdot \sqrt{\frac{\mathcal{L}_{equiv}(\theta)}{\mathcal{L}_{mean}(\theta)^\alpha}} \tag{30}$$

*Proof.* By Kurdyka (1998); Dereich & Kassing (2021), any real-analytic function $f$ satisfies the Kurdyka-Łojasiewicz inequality, which states that there exists a compact local neighborhood $U$ around any critical point $\theta^*$ and constants $c > 0, \alpha \in [1, 2)$ such that, for all $\theta \in U$:

$$\|\nabla f(\theta)\|^2 \geq c|f(\theta) - f(\theta^*)|^\alpha \tag{31}$$

When applied to the real-analytic function $\mathcal{L}_{\mathrm{mean}}$ at the global minimum $\theta^*$ with $\mathcal{L}_{\mathrm{mean}}(\theta^*) = 0$, the Kurdyka-Łojasiewicz inequality states that there exists a neighborhood $U$ around $\theta^*$ and constants $c > 0$ and $\alpha \in [1, 2)$ such that:

$$\|\nabla \mathcal{L}_{\mathrm{mean}}(\theta)\|^2 \geq c \cdot \mathcal{L}_{\mathrm{mean}}(\theta)^\alpha \tag{32}$$

Finally, by analyticity of the network and loss functions and applying lemma 7 relative to the optima $\theta^*$ with $\mathcal{L}_{\mathrm{equiv}}(\theta^*) = 0$, there exists a finite $M_{\mathcal{L}_{\mathrm{equiv}}(\theta)}(U)$ such that $\mathcal{L}_{\mathrm{equiv}}$ is $M_{\mathcal{L}_{\mathrm{equiv}}(\theta)}(U)$-smooth on the compact neighborhood $U$:

$$\|\nabla \mathcal{L}_{\mathrm{equiv}}(\theta)\|^2 \leq 2M_{\mathcal{L}_{\mathrm{equiv}}(\theta)}(U) \cdot \mathcal{L}_{\mathrm{equiv}}(\theta) \tag{33}$$

The final result follows algebraically by combining inequalities 32 and 33. $\square$

### C.5   Proof of Proposition 4

**Theorem.** *For the EScAIP architecture trained with mean-squared error loss on a non-degenerate dataset, for any fixed set of upstream parameters $\theta \setminus \boldsymbol{W}$, there exist positive constants $0 < \lambda_{min} \leq \lambda_{max}$ such that:*

$$\lambda_{min} \cdot \|\boldsymbol{W}_{\mathcal{E}\perp}\|_F^2 \leq \mathcal{L}_{equiv}(\theta) \leq \lambda_{max} \cdot \|\boldsymbol{W}_{\mathcal{E}\perp}\|_F^2 \tag{34}$$

**Remarks.** The constants $\lambda_{\mathrm{min}}$ and $\lambda_{\mathrm{max}}$ depend on the model architecture, data distribution, and other parameters $\theta \setminus \boldsymbol{W}$.

*Proof.* For a molecule $x$, the $k$-th component of the predicted force vector decomposes into a sum of contributions from $\mathbf{W}_{\mathcal{E}}$ and $\boldsymbol{W}_{\mathcal{E}\perp}$:

$$o_k(x; \boldsymbol{W}) = \underbrace{\sum_{e \in E} e_k \cdot (\bar{\boldsymbol{w}}^T \mathbf{h}(\mathbf{e}))}_{o_{eq,k}(x; \mathbf{W}_{\mathcal{E}})} + \underbrace{\sum_{e \in E} e_k \cdot (\boldsymbol{d}_k^T \mathbf{h}(\mathbf{e}))}_{\Delta o_k(x; \boldsymbol{W}_{\mathcal{E}\perp})} \tag{35}$$

where the final hidden representation $\mathbf{h}$ depends on $\theta \setminus \boldsymbol{W}$, the set of upstream parameters. Recall the equivariance error from Proposition 1, and observe that the variance of $o_k = o_{eq} + \Delta o_k$ depends only on $\Delta o_k$, as $o_{eq}$ is equivariant by construction. Thus, the equivariance error of the entire model, for a fixed set of upstream parameters and expressed as a function of the force prediction head parameters, is:

$$\mathcal{L}_{\text{equiv}}(\theta) = \mathbb{E}_{x,T} \left[ \| \Delta o(Tx; \mathbf{W}_{\mathcal{E}\perp}) - \mathbb{E}_{T'}[\Delta o(T'x; \mathbf{W}_{\mathcal{E}\perp})] \|^2 \right]$$

Now, let us denote: $g(T, x, \mathbf{W}_{\mathcal{E}\perp}) = T^{-1} \Delta o(Tx; \mathbf{W}_{\mathcal{E}\perp})$. Observe that this function $g$ is linear in our deviation parameters $\mathbf{W}_{\mathcal{E}\perp}$. By vectorizing the $h \times 3$ parameter matrix $\mathbf{W}_{\mathcal{E}\perp}$ into a $3h \times 1$ column vector $\boldsymbol{p} = \text{vec}(\mathbf{W}_{\mathcal{E}\perp})$, we can express this linear relationship as a matrix-vector product, for some matrix $\boldsymbol{M}_{T,x}$ with shape $3 \times 3h$: $g(T, x, \mathbf{W}_{\mathcal{E}\perp}) = \boldsymbol{M}_{T,x}\boldsymbol{p}$. Similarly, the rotation-averaged prediction $\bar{g}(x; \mathbf{W}_{\mathcal{E}\perp}) = \mathbb{E}_T[g(T, x, \mathbf{W}_{\mathcal{E}\perp})]$ is also a linear function, so we associate it with the matrix $\bar{\boldsymbol{M}}_x$. The equivariance error term with these linear matrix forms is:

$$\mathbb{E}_{x,T}[\|g(T, x, \mathbf{W}_{\mathcal{E}\perp}) - \bar{g}(x, \mathbf{W}_{\mathcal{E}\perp})\|^2] = \boldsymbol{p}^{\mathsf{T}} \bar{\mathcal{Q}} \boldsymbol{p} \tag{36}$$

where the matrix $\bar{\mathcal{Q}} = \mathbb{E}_{x,T}[(\boldsymbol{M}_{T,x} - \bar{\boldsymbol{M}}_x)^{\mathsf{T}}(\boldsymbol{M}_{T,x} - \bar{\boldsymbol{M}}_x)]$. Finally, observe that $\bar{\mathcal{Q}}$ is positive definite, as the as equivariance error is strictly positive on a non-degenerate dataset whenever $\mathbf{W}_{\mathcal{E}\perp} \neq \mathbf{0}$. By the properties of a positive definite matrix, the quadratic form $\boldsymbol{p}^{\mathsf{T}} \bar{\mathcal{Q}} \boldsymbol{p}$ is lower-bounded by the smallest eigenvalue of $\bar{\mathcal{Q}}$, denoted $\lambda_{min}(\bar{\mathcal{Q}})$, which is positive. It is also upper bounded by the largest eigenvalue $\lambda_{max}(\bar{\mathcal{Q}})$. This establishes the quadratic relationship on the equivariance loss as stated in the theorem.

$\square$

### C.6   Proof of Proposition 5

**Theorem.** *For any neural network whose parameters can be expressed as $\theta = \theta_{\mathcal{E}} + \theta_{\mathcal{E}\perp}$ with $\theta_{\mathcal{E}} \in \mathcal{E}$ and $\theta_{\mathcal{E}\perp} \in \mathcal{E}\perp$, and for equivariance error $\mathcal{L}_{equiv}$ defined by the variance of the output with respect to transformations, there exist positive constants $0 < \lambda_{min} \leq \lambda_{max}$ such that for a non-degenerate dataset, using $\|\cdot\|$ to denote $L_2$-norm:*

$$\lambda_{min} \|\theta_{\mathcal{E}\perp}\|^2 + \mathcal{O}(\|\theta_{\mathcal{E}\perp}\|^3) \leq \mathcal{L}_{equiv}(\theta) \leq \lambda_{max} \|\theta_{\mathcal{E}\perp}\|^2 + \mathcal{O}(\|\theta_{\mathcal{E}\perp}\|^3) \tag{37}$$

*Proof.* Applying a Taylor expansion to $\mathcal{L}_{\text{equiv}}(\theta)$ for the neural net $f$ on an input $x$ around equivariant parameters $\theta_{\mathcal{E}}$, we have:

$$f(x; \theta_{\mathcal{E}} + \theta_{\mathcal{E}\perp}) = f(x; \theta_{\mathcal{E}}) + J_{\theta_{\mathcal{E}\perp}} f(x; \theta_{\mathcal{E}}) \theta_{\mathcal{E}\perp} + \mathcal{O}(\|\theta_{\mathcal{E}\perp}\|^2) \tag{38}$$

where $J_{\theta_{\mathcal{E}\perp}} f(x; \theta_{\mathcal{E}})$ is the Jacobian of the network output with respect to parameter components $\theta_{\mathcal{E}\perp}$, evaluated at $\theta_{\mathcal{E}}$. As before, the term $f(x; \theta_{\mathcal{E}})$ is equivariant by construction, and thus drops out of the equivariance error term. The term $J_{\theta_{\mathcal{E}\perp}} f(x; \theta_{\mathcal{E}}) \theta_{\mathcal{E}\perp}$ is linear in $\theta_{\mathcal{E}\perp}$, which creates a quadratic dependence on $\theta_{\mathcal{E}\perp}$ in the variance term in $\mathcal{L}_{\text{equiv}}$.

The deviation from the twirled mean is the difference between the canonicalized prediction and its average over transformations. Let's expand this difference:

$$(T^{-1} \circ f \circ T)(x; \theta) - \mu(x; \theta) = (T^{-1} \circ f \circ T)(x; \theta) - \mathbb{E}_{T'}[(T'^{-1} \circ f \circ T')(x; \theta)] \tag{39}$$

Substituting the Taylor series and using the equivariance of $f(x; \theta_{\mathcal{E}})$:

$$= \left( f(x; \theta_{\mathcal{E}}) + [T^{-1} J \theta_{\mathcal{E}\perp} f(T(x); \theta_{\mathcal{E}})] \theta_{\mathcal{E}\perp} + \mathcal{O}(|\theta_{\mathcal{E}\perp}|^2) \right)$$
$$- \mathbb{E}_{T'} \left[ f(x; \theta_{\mathcal{E}}) + [T'^{-1} J \theta_{\mathcal{E}\perp} f(T'(x); \theta_{\mathcal{E}})] \theta_{\mathcal{E}\perp} + \mathcal{O}(|\theta_{\mathcal{E}\perp}|^2) \right] \tag{40}$$
$$= \left( T^{-1} J_{\theta_{\mathcal{E}\perp}} f(T(x); \theta_{\mathcal{E}}) - \mathbb{E}_{T'}[T'^{-1} J \theta_{\mathcal{E}\perp} f(T'(x); \theta_{\mathcal{E}})] \right) \theta_{\mathcal{E}\perp} + \mathcal{O}(|\theta_{\mathcal{E}\perp}|^2) \tag{41}$$

Let $\Delta J_{x,T} \triangleq T^{-1} J_{\theta_{\mathcal{E}\perp}} f(T(x); \theta_{\mathcal{E}}) - \mathbb{E}_{T'}[T'^{-1} J_{\theta_{\mathcal{E}\perp}} f(T'(x); \theta_{\mathcal{E}})]$. The expression becomes $\Delta J_{x,T} \cdot \theta_{\mathcal{E}\perp} + \mathcal{O}(\|\theta_{\mathcal{E}\perp}\|^2)$.

$$\mathcal{L}_{\text{equiv}}(\theta) = \frac{1}{D} \mathbb{E}_{x,T} \left[ |\Delta J_{x,T} \cdot \theta_{\mathcal{E}\perp} + \mathcal{O}(|\theta_{\mathcal{E}\perp}|^2)|^2 \right] \tag{42}$$

$$= \frac{1}{D} \mathbb{E}_{x,T} \left[ |\Delta J_{x,T} \cdot \theta_{\mathcal{E}\perp}|^2 + 2(\Delta J_{x,T} \cdot \theta_{\mathcal{E}\perp})^T \mathcal{O}(|\theta_{\mathcal{E}\perp}|^2) + |\mathcal{O}(|\theta_{\mathcal{E}\perp}|^2)|^2 \right] \tag{43}$$

The orders of the terms are:

- $\|\Delta J_{x,T} \cdot \theta_{\mathcal{E}\perp}\|^2$ is $\mathcal{O}(\|\theta_{\mathcal{E}\perp}\|^2)$.
- The cross-term is $\mathcal{O}(\|\theta_{\mathcal{E}\perp}\|) \cdot \mathcal{O}(\|\theta_{\mathcal{E}\perp}\|^2) = \mathcal{O}(\|\theta_{\mathcal{E}\perp}\|^3)$.
- The final term is $(\mathcal{O}(\|\theta_{\mathcal{E}\perp}\|^2))^2 = \mathcal{O}(\|\theta_{\mathcal{E}\perp}\|^4)$.

We will study the leading term, which is quadratic in $\theta_{\mathcal{E}\perp}$, and subsume the remainder into $\mathcal{O}(\|\theta_{\mathcal{E}\perp}\|^3)$. As $\Delta J_{x,T}$ is a linear function, we can define a matrix $\bar{\mathcal{Q}}$ that represents the averaged outer product of the Jacobian deviations: $\bar{\mathcal{Q}} \triangleq \frac{1}{D} \mathbb{E}_{x,T} [(\Delta J_{x,T})^{\mathsf{T}} (\Delta J_{x,T})]$. The equivariance error can be expressed concisely:

$$\mathcal{L}_{\text{equiv}}(\theta_{\mathcal{E}} + \theta_{\mathcal{E}\perp}) \approx \theta_{\mathcal{E}\perp}^T \bar{\mathcal{Q}} \theta_{\mathcal{E}\perp} \tag{44}$$

The matrix $\bar{\mathcal{Q}}$ is positive definite for a non-degenerate dataset when $\theta_{\mathcal{E}\perp} \neq 0$. Using the Rayleigh-Ritz theorem, this quadratic form is thus bounded by the smallest and largest eigenvalues:

$$\lambda_{\min} \|\theta_{\mathcal{E}\perp}\|_2^2 \leq \theta_{\mathcal{E}\perp}^T \bar{\mathcal{Q}} \theta_{\mathcal{E}\perp} \leq \lambda_{\max} \|\theta_{\mathcal{E}\perp}\|_2^2$$

Reincorporating the remainder term in our Taylor expression, we arrive at:

$$\lambda_{\min} |\theta_{\mathcal{E}\perp}|_2^2 + \mathcal{O}(|\theta_{\mathcal{E}\perp}|_2^3) \leq \mathcal{L}_{\text{equiv}}(\theta) \leq \lambda_{\max} |\theta_{\mathcal{E}\perp}|_2^2 + \mathcal{O}(|\theta_{\mathcal{E}\perp}|_2^3) \tag{45}$$

$\square$

## C.7 Proof of Proposition 6

**Theorem.** *Under the same conditions as the Taylor expansion theorem above, the norm of the gradient of the equivariance loss with respect to the non-equivariant parameters is bounded by the deviation itself. Specifically, there exists a constant $C$ such that:*

$$\|\nabla_{\theta_{\mathcal{E}\perp}} \mathcal{L}_{equiv}(\theta)\| \leq C \cdot \|\theta_{\mathcal{E}\perp}\|$$

*Proof.* From our proof in C.6, we know $\mathcal{L}_{\text{equiv}}(\theta) \approx \boldsymbol{p}^{\mathsf{T}} \bar{\mathcal{Q}} \boldsymbol{p}$, where $\boldsymbol{p} = \text{vec}(\theta_{\mathcal{E}\perp})$. The gradient of a quadratic form is linear: $\nabla_{\boldsymbol{p}} \mathcal{L}_{\text{equiv}} = 2\bar{\mathcal{Q}}\boldsymbol{p}$. Taking norms, we get $\|\nabla_{\boldsymbol{p}} \mathcal{L}_{\text{equiv}}\| = \|2\bar{\mathcal{Q}}\boldsymbol{p}\| \leq 2\|\bar{\mathcal{Q}}\|\|\boldsymbol{p}\|$. Setting $C = 2\lambda_{max}$ or $2\|\bar{\mathcal{Q}}\|_2$ gives the result. $\square$

# D Methods & Experimental Details

## D.1 EScAIP

We trained EScAIP 6M on a subset of SPICE with 950k training examples used by Qu & Krishnapriyan (2024) for 30 epochs with batch size 64. SPICE is a dataset with of small molecule 3D conformers with energies and forces computed by quantum-mechanical density functional theory (Eastman et al., 2024). We varied model size from 1M, 4M and 6M, varied training set size from 950k, 50k, 5k, and 500 (with batch size 1), and varied the optimizer or learning rate. The model predicts a 3D force vector for each atom based on density functional theory, mapping an input molecule with $N$ atoms to an output in $\mathbb{R}^{3N}$. This task is physically equivariant to the special orthogonal group $SO(3)$ acting on atom coordinates in $\mathbb{R}^3$.

We follow the same training recipe as the original repository, which does not use data augmentation. We suspect that data augmentation is not as important for EScAIP because it operates on rotation-invariant features.

For further details and configuration files, please refer to our code repository.

## D.2 Proteína

We trained Proteína at 60M without triangular attention and 400M with triangular attention on the full Protein databank (PDB) dataset with 225k training examples. We also trained models on 1% of the PDB with 2k examples and 0.1% with 200 examples. Flow matching trains a model jointly over $t$, flow matching time, ranging from $t = 0$ for noise and $t = 1$ for data. We measure metrics at $t = 0, 0.2, 0.4, 0.6, 0.8, 0.9, 0.95$, and 0.99, and use red colors for high $t$ close to the data, and blue-purple colors for low $t$ near noise in Figure 3. The model learns to approximate the velocity field of a probability flow that transforms random noise into structured protein backbones. For a molecule with $N$ alpha carbon atoms, the network maps noised atom coordinates and a time $t \in [0, 1]$ to a velocity vector in $\mathbb{R}^{3N}$. The learning task is made rotationally equivariant through data augmentation, aligning it with $SO(3)$ acting on atom coordinates in $\mathbb{R}^3$.

For further details and configuration files, please refer to our code repository.

## D.3 VoxMol

Following Pinheiro et al. (2023), we represent each molecule using a 3D voxel grid by placing a continuous Gaussian density at each atom's position. Each atom type is assigned a distinct input channel, producing a 4D tensor of shape $[c \times l \times l \times l]$, where $c$ denotes the number of atom types and $l$ is the edge length of the voxel grid. The voxel values are normalized between 0 and 1.

The denoising task arises from the use of walk-jump sampling for generating molecules (Saremi & Hyvärinen, 2019). This uses a two-step score-based sampling method. The "walk" phase involves running $k$ steps of Langevin Markov chain Monte Carlo on a randomly initialized noisy voxel grid, simulating a stochastic trajectory along a manifold. The "jump" phase applies a denoising autoencoder (DAE) to clean up the noisy sample using a forward pass of the trained model at step $k$. The DAE is trained on voxelized molecules corrupted with isotropic Gaussian noise, with a mean squared error (MSE) loss between prediction and ground truth. WJS provides a fast alternative to diffusion models by requiring only a single noise and denoise step (Pinheiro et al., 2023; Nowara et al., 2024).

**Architecture** The VoxMol architecture is based on a 3D U-Net with convolutional layers spanning four resolution scales, and includes self-attention modules at the two coarsest levels Pinheiro et al. (2023). During training, data augmentation is performed by applying random rotations and translations to each sample. For further architectural and training details, refer to Pinheiro et al. (2023).

**Measuring whether latent representations learn to respect equivariance** To evaluate whether VoxMol learns equivariant latent features, we analyze cosine similarity between latent embeddings under two scenarios.

First, we examine representations of the *same molecule under rotation*. Let $\mathbf{x}$ be a molecule and $R_k$ a discrete rotation operator (e.g., 90° around an axis). Using the encoder $\phi(\cdot) \in \mathbb{R}^{C \times D \times H \times W}$, with $C = 512$ and spatial dimensions $8 \times 8 \times 8$, we define the spatially pooled latent vector:

$$\bar{\phi}(\mathbf{x}) = \frac{1}{DHW} \sum_{d,h,w} \phi(\mathbf{x})[:, d, h, w]$$

We then compute:

$$\text{sim}_{\text{same}} = \cos\left(\bar{\phi}(R_k(\mathbf{x})),\ R_k(\bar{\phi}(\mathbf{x}))\right)$$

This measures whether encoding a rotated molecule is equivalent to rotating the latent vector of the original input—a key signature of learned equivariance.

Second, to obtain a baseline, we compute cosine similarities between embeddings of *randomly selected different molecules*:

$$\text{sim}_{\text{diff}} = \cos\left(\bar{\phi}(\mathbf{x}_i),\ \bar{\phi}(\mathbf{x}_j)\right), \quad \text{with } \mathbf{x}_i \neq \mathbf{x}_j$$

We compute these metrics across 1000 molecules for various rotation angles along all three axes. Cosine similarities are calculated over the 512-dimensional latent vectors and visualized using violin plots to capture the distributional differences in Figure 11.

**Findings.** Cosine similarity between rotated versions of the same molecule tends to decrease as rotation angle increases, reflecting imperfect latent equivariance. While same-molecule embeddings remain more similar to each other than to embeddings of different molecules, the overlap between their distributions grows with rotation. This suggests that although the encoder partially preserves geometric structure, the latent space does not fully achieve rotation equivariance, indicating potential for improved regularization or architectural design.

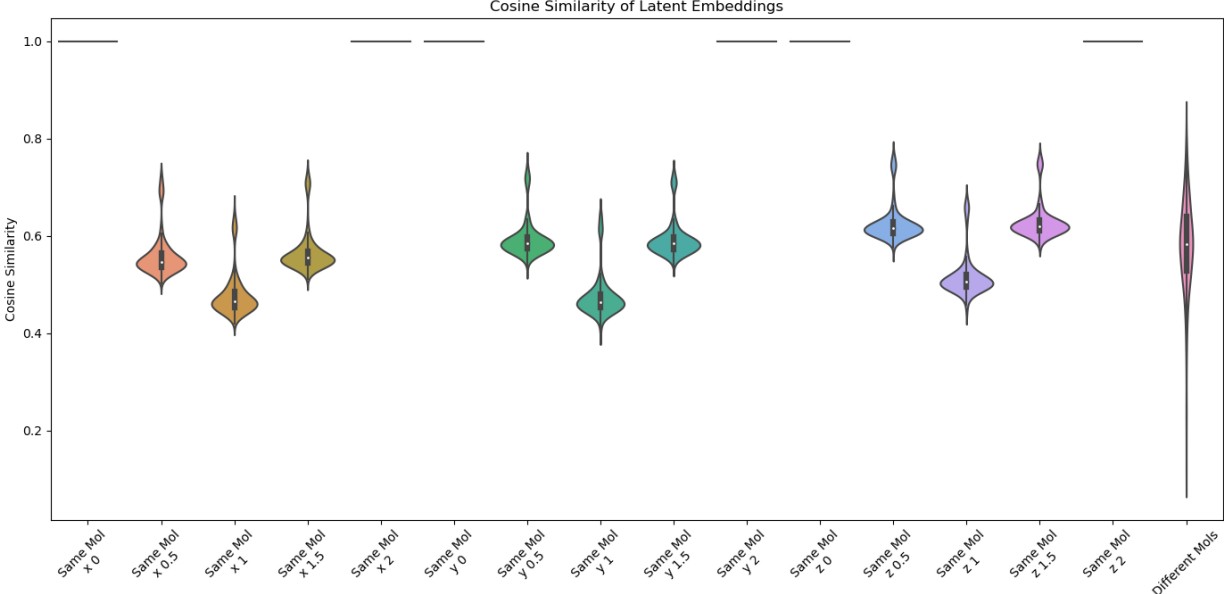

Figure 11: VoxMol: Cosine similarity of molecule latent representations with different rotations. x, y, z indicate rotation axes, and numbers 0, 0.5, 1, 1.5, 2 correspond to 0, 90, 180, 270, 360 degrees of rotation. The last column depicts cosine similarity between different molecules.

## D.4 Metrics

To compute equivariance error, twirled prediction, error, percent MSE loss from equivariance error, and gradient norms, 10 rotations per sample were used in EScAIP and Proteína. 4 rotations per sample were

used for VoxMol. These numbers were found to be sufficient to provide a stable signal for metrics which was robust to randomness and resampling (§B.1). Each measurement point uses a different set of random rotations, so each metric's stability over time also reflects the stability of our measurements. For EScAIP, these metrics were computed on the first four (fixed) validation batches with batch size of 16, for a total of 64 samples. For Proteína, these metrics were computed on the first eight (fixed) validation batches with batch size of 3, for a total of 24 samples. The total MSE loss on these subsets was indicative of the total validation MSE loss, indicating these sample sizes were sufficient to provide a stable and representative signal for these metrics.

### D.5 Hessian Analysis and Condition Numbers

To plot the loss landscape, we selected a subset of parameters in each architecture. For EScAIP, we used the final FFN (with a non-linearity) and the final linear head, for a combined total of 33k parameters. For Proteína, we used the final linear head with 1.5k parameters. We computed the Hessian of this parameter subset for the total MSE loss using one fixed training batch with ten rotations. We then performed eigendecomposition of the total MSE loss Hessian to find the eigenvectors for the largest positive eigenvalue, and minimum positive eigenvalue, which formed the two axes for plotting the loss landscape. We selected a step size approximately 2-3x the training step size at that checkpoint, which is estimated by multiplying the training learning rate with the total parameter gradient norm at that checkpoint. We then create a 2D grid of perturbations to the parameter subset, and compute $\mathcal{L}_{\mathrm{mean}}$ and $\mathcal{L}_{\mathrm{equiv}}$ at each point on the grid. Importantly, the axes and the step size are the same for both $\mathcal{L}_{\mathrm{mean}}$ and $\mathcal{L}_{\mathrm{equiv}}$.

To compute the condition numbers, we computed the Hessian of the same parameter subsets for $\mathcal{L}_{\mathrm{mean}}$ and $\mathcal{L}_{\mathrm{equiv}}$ separately, and performed eigendecomposition on them separately. We reported the condition number as the ratio between the largest positive eigenvalue and the minimum positive eigenvalue.

Here, we provide figures of the condition numbers across 20 minibatches.

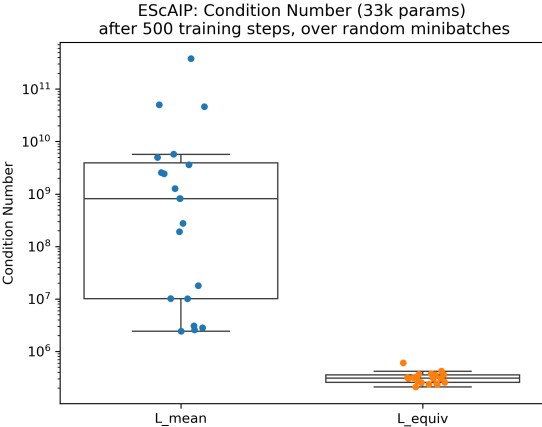

Figure 12: EScAIP: Condition numbers across 20 minibatches.

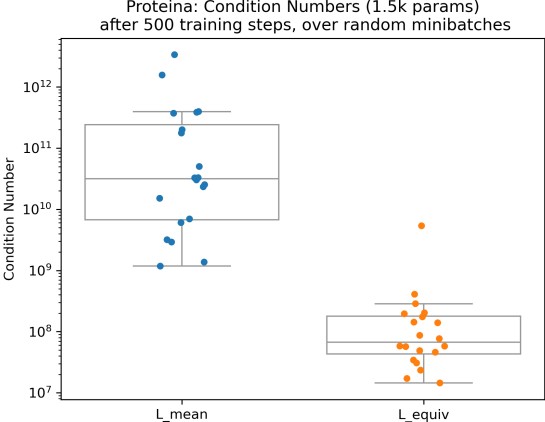

Figure 13: Proteína: Condition numbers across 20 minibatches.

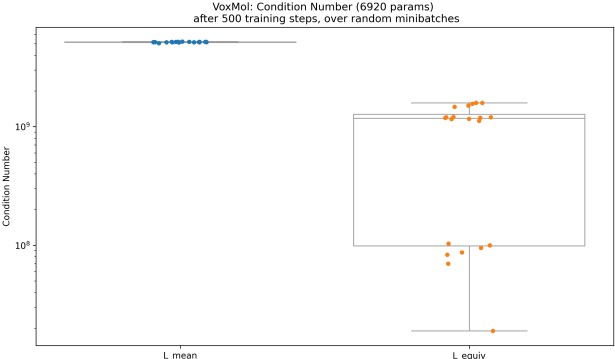

Figure 14: VoxMol: Condition numbers across 20 minibatches.

## D.6 Bias-corrected finite sample estimators

In this section, we discuss unbiased finite sample estimators for $\mathcal{L}_{\mathrm{mean}}$ and $\mathcal{L}_{\mathrm{equiv}}$, which can be used to build a bias-corrected finite sample estimator for the percent loss from equivariance error. Suppose we have $N > 1$ finite group samples. Denote the the exact group-averaged mean as $\mu$ (i.e., the exact expectation, which could be calculated with infinite group samples), and our finite sample estimate of the mean as $\hat{\mu}$.

$$\mu(x) = \mathbb{E}_T \left[ (T^{-1} \circ f \circ T)(x) \right] \tag{46}$$

$$\hat{\mu}(x) = \frac{1}{N} \sum_{i=1}^{N} \left[ (T_i^{-1} \circ f \circ T_i)(x) \right] \tag{47}$$

Similarly, denote $\sigma^2$ as the exact variance over the group. Denote $\hat{\sigma}^2$ as our biased finite sample estimate of the variance (biased because it divides by $N$, not $N-1$):

$$\hat{\sigma}^2 = \frac{1}{N} \sum_{i=1}^{N} \left( (T_i^{-1} \circ f \circ T_i)(x) - \hat{\mu}(x) \right)^2 \tag{48}$$

$$\mathbb{E}_T[\hat{\sigma}^2] = \frac{N-1}{N} \sigma^2 \tag{49}$$

The MSE loss of our finite sample estimate $\hat{\mu}$ is biased, relative to $\mu$. For exposition, we denote $\mu = \mu(x)$.

$$\mathbb{E}_T[(\hat{\mu} - y)^2] = \underbrace{(\mu - y)^2}_{\text{True squared bias}} + \frac{\sigma^2}{N} \tag{50}$$

This can be derived as follows:

$$(\hat{\mu} - y)^2 = (\hat{\mu} + \mu - \mu - y)^2 \tag{51}$$
$$= (\hat{\mu} - \mu)^2 + 2(\mu - y)(\hat{\mu} - \mu) + (\mu - y)^2 \tag{52}$$

Now, take an expectation over group samples $T$:

$$\mathbb{E}_T[(\hat{\mu} - y)^2] = (\hat{\mu} + \mu - \mu - y)^2 \tag{53}$$
$$= \mathbb{E}_T[(\hat{\mu} - \mu)^2] + 2(\mu - y)\underbrace{\mathbb{E}_T[(\hat{\mu} - \mu)]}_{} + \mathbb{E}_T[(\mu - y)^2] \tag{54}$$
$$= \underbrace{\mathbb{E}_T[(\hat{\mu} - \mu)^2]}_{=\sigma^2/N} + (\mu - y)^2 \tag{55}$$

Thus, unbiased finite sample estimators are:

$$\mathbb{E}_T\left[(\hat{\mu} - y)^2 - \frac{1}{N-1}\hat{\sigma}^2\right] = (\mu - y)^2 \tag{56}$$
$$\mathbb{E}_T\left[\frac{N}{N-1}\hat{\sigma}^2\right] = \sigma^2 \tag{57}$$

**Proposition 8.** *The finite sample estimator for the percent loss from equivariance error*

$$\frac{N}{N-1}\frac{\hat{\sigma}^2}{(\hat{\mu} - y)^2 + \hat{\sigma}^2} \tag{58}$$

*has a numerator that is unbiased to $\sigma^2$, and has a denominator is unbiased to $(\mu - y)^2 + \sigma^2$.*

*Proof.* For the numerator, see equation 48. For the denominator: $\mathbb{E}_T[(\hat{\mu} - y)^2 + \hat{\sigma}^2] = (\mu - y)^2 + \sigma^2$.

$$\mathbb{E}_T[(\hat{\mu} - y)^2] + \mathbb{E}_T[\hat{\sigma}^2] = (\mu - y)^2 + \frac{1}{N}\sigma^2 + \frac{N-1}{N}\sigma^2 \tag{59}$$
$$= (\mu - y)^2 + \sigma^2 \tag{60}$$

$\square$

A similar argument can be used to motivate the $N/(N-1)$ correction for general convex losses, where the correction is second-order accurate.

