# OpenReview forum: "Training Dynamics of Learning 3D-Rotational Equivariance"
_TMLR — Accepted by TMLR_

### Review · Reviewer_S3vC · 2025-09-11

**Summary Of Contributions:**

This work considers the setting of supervised learning in problems that feature equivariances. The typical choice in these problems is either to design an architecture that is inherently equivariant (but may be substantially more expensive for training and inference), or perform data augmentation with a standard architecture but penalise violations of equivariance. The work proposes to decompose the overall loss into two parts: one for the equivariance violations, and another for the main prediction task itself. The authors perform an empirical investigation on several 3D molecular prediction tasks, finding several interesting phenomena that are mostly consistent across tasks: a "dip" followed by a rise in the validation loss component, and a loss landscape for learning equivariance that is much smoother than the one for the main task. The work then derives a series of theoretical results on the learning dynamics, studying the magnitudes of the losses, their gradients, and the deviation in parameters from a reference perfectly equivariant function. The paper makes both theoretical and empirical contributions.

**Audience:**

Yes

**Audience Explanation:**

Yes, this paper is highly relevant for the TMLR audience and addresses an important problem with many applications in the sciences.

**Claims And Evidence:**

Yes

**Claims Explanation:**

Yes, both empirical and theoretical claims are appropriately supported by evidence. I lack the necessary background to check the correctness of the proofs, however.

**Requested Changes:**

The paper is very well thought out,  well-executed, and well-written. My opinion of it is positive overall. I have a few suggestions that, in my opinion, would strengthen the paper. First the more important points:

C1. The theoretical results are mostly disjoint from the empirical experiments, with the exception of those in 4.1, which do not hold consistently. Is there a way to also validate those in 4.2 and 4.3? If the answer is no, why is this challenging or impractical?

C2. Can you make any statements about how difficult it would be to adapt the decomposition for classification tasks instead of regression with the MSE objectives? Would this still allow you to study the problem empirically even if not theoretically?

C3. Do you intend to release the code publicly in the case of acceptance? Code was not submitted as supplementary material, which makes it harder to reproduce the work.

Typos and small suggestions:

- "models rapidly become nearly equivariant within 1k-10k training steps" -> seems this can vary substantially so "rapid" is sufficient for the abstract?
- Footnote on author roles irrelevant for double-blind submission.
- Bold part of point (ii) in Section 1 is an incomplete sentence.
- 3.1, first bullet point "its" -> "it is"
- 3.3, double period towards end of first paragraph
- 3.5 first paragraph: "we provide *an* analysis here"
- Section 4: can you provide a definition of gradient purity? The term is non-standard IMO.
- Check references: Nowara et al. is duplicated; format for the conferences (e.g. NeurIPS) is inconsistent

---

> ### Author Response · Authors · 2025-10-23
>
> > The paper is very well thought out, well-executed, and well-written. My opinion of it is positive overall. I have a few suggestions that, in my opinion, would strengthen the paper. First the more important points:
> > C1. The theoretical results are mostly disjoint from the empirical experiments, with the exception of those in 4.1, which do not hold consistently. Is there a way to also validate those in 4.2 and 4.3? If the answer is no, why is this challenging or impractical?
>
> We added an experimental investigation for proposition 4 on EScAIP’s force prediction head’s deviation from exactly equivariant parameter subspace, to the equivariance error, and found a Pearson correlation of 0.94 and Spearman correlation of 0.99. We report this in the main text and appendix section B.4. Furthermore, we have added context and discussions around our theoretical results to better connect their motivations, and expanded our discussion of experimental investigations on our propositions.
>
> > C2. Can you make any statements about how difficult it would be to adapt the decomposition for classification tasks instead of regression with the MSE objectives? Would this still allow you to study the problem empirically even if not theoretically?
>
> Our framework applies in a clean and simple manner to classification tasks. First, note that our framework is presented for equivariance, but it also applies to invariance - everything applies correctly, just taking T^{-1} to be the identity map for invariance. This is important to set up, as classification is an invariant task, unlike the regression tasks we consider in our primary manuscript which are equivariant tasks.
>
> The standard objective for classification is soft-max cross entropy, which takes logits as input, and returns a scalar loss. This objective is convex in the input logits, and returns non-negative outputs, so we can apply our result that % loss from equivariance error can be computed as the normalized difference between the total loss (over data augmentations from the group) and the loss of the twirled (group-averaged) prediction. This is straightforward to compute: for a given input x, sample N rotations, predict logits for each rotated input. The twirled loss is the loss computed on the averaged logits, and the total loss is the averaged loss over the logits. The difference, divided by the total loss, is the % loss from equivariance error for soft-max cross-entropy for classification. This procedure is straightforward and is readily applicable to any symmetry group, model architecture, and classification task.
>
> > C3. Do you intend to release the code publicly in the case of acceptance? Code was not submitted as supplementary material, which makes it harder to reproduce the work.
>
> Yes, we will release the code publicly in case of acceptance. We note that our code simply adds some callbacks to existing code bases to track metrics during training.
>
> Thank you for pointing out typos and small suggestions. We have revised and fixed all of these suggested. For reference formats for conferences, we have tried to ensure that all NeurIPS citations are similar. However, citation formats can vary depending on where the bibtex is pulled from (OpenReview, arxiv, conference website). If you have more specific suggestions on standardizing formats or tips on how to do so, we would gladly welcome them.

---

### Review · Reviewer_4MxY · 2025-09-19

**Summary Of Contributions:**

• Introduces a “twist & twirl” framework to measure learned equivariance by comparing twisted vs. twirled predictions.
• Derives a principled loss decomposition under data augmentation: total loss = prediction error + equivariance error, exact for MSE. Provides an interpretable metric: % of loss due to non-equivariance.
• Empirical study on 3D rotations (force-field prediction, flow matching, voxel denoising) shows near-equivariance emerges early and robustly, with smoother optimization.
• Theoretical results connect equivariance loss to gradient alignment, parameter distance, and scaling laws.
• Practical insights: residual non-equivariance often has small penalties; non-equivariant models can be more efficient, while simple test-time twirling reduces loss.

Key Strengths
• Clear, general framework with exact decomposition under MSE.
• Strong, consistent empirical evidence across tasks and scales.
• Theory explains the fast emergence of near-equivariance and provides actionable test-time methods.

Key Weaknesses / Limitations
• Exactness limited to convex/MSE losses; analysis approximate for others.
• Test-time twirling adds overhead without full trade-off analysis.

**Audience:**

Yes

**Audience Explanation:**

• ML theory (loss decomposition, gradient dynamics).
• Representation learning/symmetry (learned equivariance metrics).
• Molecular ML practitioners (3D tasks, practical trade-offs, and twirling).

**Claims And Evidence:**

Yes

**Claims Explanation:**

• A formal loss decomposition (exact for MSE) is derived, so the “equivariance error” metric is well-defined and testable.
• Theory is tied to parameters (equivariant subspace vs. orthogonal part), with a quadratic relationship proved for EScAIP—making falsifiable predictions.
• Experiments across three distinct 3D molecular tasks show equivariance error plunges early (≈1k–10k steps) and stays small, robust to model/data/optimizer choices.
• Landscape probes indicate Lequiv is smoother/better-conditioned than Lmean, explaining the rapid convergence.
• Gradient/loss correlations support the “stochastic equivariant learning dynamics” explanation.

**Requested Changes:**

I believe the paper makes a solid contribution, and I would like the authors to perform the following to demonstrate the advantages of their work.
1. Uncertainty quantification: For “% loss from non-equivariance” curves, add CIs (bootstrap over molecules & rotations) and a sensitivity study vs. # of rotations per sample.
2. Fair baselines: Add equal-compute comparisons vs. strong equivariant models; report accuracy per GPU-hour and wall-clock; quantify test-time twirling overhead.
3. Landscape analysis: Repeat conditioning/Hessian probes across multiple batches/checkpoints/parameter blocks; report dispersion (median/IQR).
4. Scope of theory: Make MSE exactness explicit; either validate on another convex loss (e.g., MAE) or narrow claims accordingly.
5. Data rigor: Specify precise train/val/test splits; check leakage (near-duplicates); add held-out test results for all settings.

In the following, I will list the changes that would strengthen the work even further, but are not necessary:
• Parameter tracking: Track ∥θ_E⊥∥ over training and correlate with L_equiv; for EScAIP, show head weights (w_x,w_y,w_z) merging to a shared subspace.
• Latent diagnostics: Add probes for learned (non)equivariance (e.g., predict rotation/canonical frame from latents) and report AUROC over training.
• Augmentation ablations: None vs. single-axis vs. full SO(3) to isolate drivers of early equivariance.
• Broader groups: Small demo on a second symmetry (e.g., permutations) or an appendix plan with expected dynamics.
• Statistical reporting: Error bars/CIs on all time-series; include r and p-values for gradient-purity correlations.
• Practical guidance: Add a 1-page table—when to prefer equivariant vs. symmetry-agnostic (+ aug/twirling) by data scale/compute/latency constraints.
• Twirling recipe: Pseudocode + estimator variance notes; sensitivity to rotation count.
• Writing/figures: Define “twist/twirl” once with a notation table; replace “dip” with precise wording; unify EScAIP naming; enlarge axis labels; move essential metric details from appendix to main text.

---

> ### Author Response · Authors · 2025-10-23
>
> > I believe the paper makes a solid contribution, and I would like the authors to perform the following to demonstrate the advantages of their work.
> > (1) Uncertainty quantification: For “% loss from non-equivariance” curves, add CIs (bootstrap over molecules & rotations) and a sensitivity study vs. # of rotations per sample.
>
> We have added a sensitivity study of the % loss from equivariance error, vs. the number of random rotation samples, to appendix section B.1. We used 100 random 3D rotations on a model checkpoint to estimate its 4% loss from equivariance error, and report the standard error of the percent loss from equivariance error with varying number of rotation samples from 1 to 100. We find that even with ~10 rotations as used in our work, the standard error is a small fraction of the statistic’s value. Finally, we note that all metrics reported and plotted in the figures, at different training times, use different randomly sampled rotations. Our metrics’ stability across training time is indicative of the robustness of our measurement setup.
>
> > (2) Fair baselines: Add equal-compute comparisons vs. strong equivariant models; report accuracy per GPU-hour and wall-clock; quantify test-time twirling overhead.
>
> Comparisons vs. strong equivariant models: We believe this is out-of-scope for our work. We explicitly aim to not directly compare equivariant models to non-equivariant models, and instead focus specifically on the training dynamics of how non-equivariant models learn 3d-rotational equivariance (which is our paper’s title). As we write in the introduction: “While it is possible to directly compare efficiency gaps [for equivariant models] to loss penalties from imperfect symmetry [in non-equivariant models], this is easily confounded by implementation details. To provide a more fundamental insight, we instead isolate and quantify a key source of potential underperformance in symmetry-agnostic models.” We believe this focus provides a more fundamental insight, as the learning dynamics of non-equivariant models is more likely to be a persistent phenomenon than the efficiency gap of equivariant models, which is more readily impacted by engineering efforts. We believe that our work does a thorough job within our specific focus, and that our conclusions are interesting and relevant to the research community. As such, we propose to leave equal-compute comparisons vs. strong equivariant models for future work.
>
> Test-time twirling overhead: We remark this as a possibility to highlight how simple it is to resolve non-equivariance, but in the context of our manuscript, we do not believe that test-time twirling is necessary because the models we studied are already nearly equivariant from training. Quantitatively, twirling with N rotations incurs an N-times overhead on test-time predictions, though this is easily batched and processed in parallel.
>
> > (3) Landscape analysis: Repeat conditioning/Hessian probes across multiple batches/checkpoints/parameter blocks; report dispersion (median/IQR).
>
> We have added new figures, repeating our condition number calculations across 20 minibatches for L_mean and L_equiv for all three models, to the appendix (Figs. 12-14), and updated section 3.4. Our new figures show that the condition numbers, and our findings that L_equiv is better conditioned than L_mean, are robust to random minibatches. The primary focus of our Hessian analysis is studying the early-training phase to understand why % equivariance error diminishes rapidly; as such, we continue to focus on the same checkpoints at early training.
>
> > (4) Scope of theory: Make MSE exactness explicit; either validate on another convex loss (e.g., MAE) or narrow claims accordingly.
>
> We have narrowed our claims in the introduction to explicitly state that we conduct our analysis for mean-squared error. Furthermore, in our theory section (4), we have amended several propositions to support general convex losses. In each proposition, we clearly state whether general convex losses or MSE loss is assumed.
>
> > (5) Data rigor: Specify precise train/val/test splits; check leakage (near-duplicates); add held-out test results for all settings.
>
> We use the same train/val/test splits as prior work, which we believe maximizes relevance and makes it easier for researchers to interpret and contextualize our results. Furthermore, improved model performance is not a key claim of our work; if it was, it could be more appropriate to consider checking leakage. For some settings, we use validation data to measure statistics on data unseen during training. We do not use validation data for model selection or for any type of optimization.

---

> ### Author Response · Authors · 2025-10-23
>
> > In the following, I will list the changes that would strengthen the work even further, but are not necessary:
> > • Parameter tracking: Track ∥θ_E⊥∥ over training and correlate with L_equiv; for EScAIP, show head weights (w_x,w_y,w_z) merging to a shared subspace.
>
> We added an experimental investigation for proposition 4 on EScAIP’s force prediction head’s deviation from exactly equivariant parameter subspace, to the equivariance error, and found a Pearson correlation of 0.94 and Spearman correlation of 0.99. We report this in the main text and appendix section B.4.
>
> > • Statistical reporting: Error bars/CIs on all time-series; include r and p-values for gradient-purity correlations.
>
> Our gradient purity correlation plots in the appendix include r and p-values.

---

### Review · Reviewer_Rsqg · 2025-10-14

**Summary Of Contributions:**

This work investigated the training dynamics of learning 3D equivariance by decomposing the loss into a twisted prediction error and equivariance error. A few interesting observations were found in terms of fast learning of equivariance. The authors also tried to provide a theoretical justification.

**Audience:**

Yes

**Audience Explanation:**

Achieving equivariance is an important task for machine learning, especially for non-equivariant models with data-augmentation.

**Claims And Evidence:**

No

**Claims Explanation:**

The authors provide a comprehensive numerical studies of fast convergence of equivariance error. However, I found most of their efforts in providing theoretical justifications questionable. See more details in my comments below.

**Requested Changes:**

Major issues:

The loss decomposition in 2.1 is formulated for a prediction test. It is not clear how it generalizes for training objectives of generative tasks like the conditional flow-matching loss for Proteina. The author should also discuss the interpreation of the loss for non-predictions tasks in section 3.2 and 3.3.

Section 3.2, it is not clear how to interpret the traning dynamics for various flow time t, and why the loss with respect to t is interesting to the audience. In particular, the training of flow models is simultaneous for all t as a whole, and given that the sampling also requires using the velocity field at all times, why is the point of investigating the training dynamics of at a particular flow time t? The author should add discussion on the interpretation of these quanties that are numerically tested.
Unfortunately, the author reports observations of the percentage validation loss for various but no conclusion/interpretation is given.

Figure 3, it seems that the percentage equivariance error is decaying fast for noise regime (t close to zero). Is this because a Gaussian noise was used as it is rotation invariant automatically?

It is claimed in Section 4.1 that, as \epsilon shrinks, \nabla L_mean will dominate \nabla_L. This is certainly not true in general because small loss does not imply small gradient. The author tried to prove this argument via the M-smooth assumption so that the gradient of \epsilon is bounded by itself. However, this is not a mild condition and it is not true in general. The author should either provide insights why such condition can be met for certain models or at least numerically verify it. There is a paragraph discussing numerical validation but it is not clear why M-smoothness holds.

The proof to Proposition 3 is based on several previous results and mathematical notions that are not defined in the paper, making it challenging to read and understand the proof. The author should at least define important concepts (like KL inequality), and prove all claims such as analyticity implies M-smoothn ess. I suspect these claims are not true in general, for example x^4 is analytic but not M-smooth on the real line.

Section 4.2, the decomposition here only makes sense when the equivariant models form a linear subspace.  The author tries to follow Nordenfors et al in Appendix A.1. However, this claim only holds for linear models. The author claims that ``element-wise applied nonlinearities are equivariant”, but this is incorrect for most activation functions. For example, ReLU is not rotation equivariant.

Minor issues:

Page 4: the author should (briefly) explain that the first term in the decomposition of the loss is averaged over T.

Fonts in some of figures (Figure 3) are too small to read.

The M_\epsilon-smooth condition is never defined in the main text. There also seems to be a typo in its definition in the Proof of Proposition 2.

Typos:

Page 1, 2nd paragraph missing period.

Page 3, extra period in the first sentence in section 3.3.

Page 20, proposition 3 is labeled as proposition 7.


Notations:

The ordering in the legend of Figure 2f should be fixed.

Figure 4c, text should be added as in 4b to distinguish the prediction error and equivariance error.

---

> ### Author Response · Authors · 2025-10-23
>
> Thank you for your review. We are glad that you found our empirical observations to be interesting. We have responded to your comments below, and have edited the manuscript to dial back the claims around the theoretical investigations.
>
> > The loss decomposition in 2.1 is formulated for a prediction test. It is not clear how it generalizes for training objectives of generative tasks like the conditional flow-matching loss for Proteina. The author should also discuss the interpreation of the loss for non-predictions tasks in section 3.2 and 3.3.
>
> The conditional flow-matching training objective for Proteina is a mean-squared error regression objective at each flow-matching time t. The loss decomposition in 2.1 is formulated for mean-squared error regression, so it applies directly to Proteina. In general, prediction tasks as well as generative tasks can use MSE objectives. VoxMol’s autoencoding task is trained with an MSE loss as well, which also can be viewed as training a generative model when paired with a sampler on the latents.
>
> In Proteina’s flow matching task, equivariance means the model would output the same yet rotated velocity, when the input noised molecule rotates. Such equivariance is a common desired property for molecular generative models (Proteina, AlphaFold3), thus motivating equivariant generative model architectures, or training generative models with rotation data augmentation.
>
> In VoxMol’s autoencoding task, equivariance means the model would output a rotated predicted reconstruction when the input molecule rotates; this is a commonly desired property when using the decoder as a generative model.
>
> We have added these motivations and interpretations to sections 3, 3.2 and 3.3.
>
> > Section 3.2, it is not clear how to interpret the traning dynamics for various flow time t, and why the loss with respect to t is interesting to the audience. In particular, the training of flow models is simultaneous for all t as a whole, and given that the sampling also requires using the velocity field at all times, why is the point of investigating the training dynamics of at a particular flow time t? The author should add discussion on the interpretation of these quanties that are numerically tested. Unfortunately, the author reports observations of the percentage validation loss for various but no conclusion/interpretation is given.
>
> The loss with respect to t is an important object in diffusion and flow matching, as it directly reflects the quality of the learned velocity field, relative to the ideal target which is the loss-optimal velocity field. Sampling indeed requires using the velocity field at all times; thus, non-equivariant sampling is caused by non-equivariant velocity fields (at some or all t).
>
> Our work provides a way to quantify the non-equivariance of velocity fields at any time t, which directly relates to whether overall sampling is equivariant or not. Our work additionally provides actionable insights, improving on prior work. As we write in the related works section (paraphrased): “Vonessen et al. (2025) study equivariance in flow matching, but their measure of equivariance conflates task difficulty, which gets easier as t → 1, with equivariance error. We correct for this issue, and find that t = 0.9 is the most problematic time for non-equivariance, whereas they find t = 0.5 instead.”
>
> If one is interested in improving equivariance of sampling from Proteina, our results specifically point to t=0.9 for additional data augmentation or test-time equivariant tactics, a finding that prior work has not found.
>
> Finally, we acknowledge that training or held-out loss, even if critical for training and monitoring, are often secondary to generative quality metrics, which researchers and end-users may care more about – reducing loss is often necessary, but may not be sufficient, for high generative quality. While pursuing an understanding of equivariant architectures vs. non-equivariant ones on final generative quality is of substantial research interest, we view it as a challenging research problem that is difficult to tackle directly. Our paper tackles this using a more tractable, adjacent approach by studying held-out loss.
>
> We hope this highlights why quantifying equivariance error at different times t in flow matching is relevant to the research community.
>
> > Figure 3, it seems that the percentage equivariance error is decaying fast for noise regime (t close to zero). Is this because a Gaussian noise was used as it is rotation invariant automatically?
>
> Yes, Proteina uses a Gaussian source distribution. We believe that the Gaussian distribution’s rotation invariance is a strong explanation for why equivariance is easier to learn at earlier t (~0).

---

> > ### Author Response · Authors · 2025-10-23
> >
> > > It is claimed in Section 4.1 that, as \epsilon shrinks, \nabla L_mean will dominate \nabla_L. This is certainly not true in general because small loss does not imply small gradient. The author tried to prove this argument via the M-smooth assumption so that the gradient of \epsilon is bounded by itself. However, this is not a mild condition and it is not true in general. The author should either provide insights why such condition can be met for certain models or at least numerically verify it. There is a paragraph discussing numerical validation but it is not clear why M-smoothness holds.
> >
> > Thank you for bringing our attention to this. Previously, our first proposition in section 4.1 considered a global smoothness assumption, which was overly restrictive and not very realistic for neural networks. We have revised and updated the proposition to work on weaker, more realistic, and more precise assumptions. Specifically, the proposition now works off of an assumption that the neural network and losses are analytic, which is commonly satisfied in practice. By a standard real analysis result, analytic functions are smooth on compact subsets of their domain (for a neural net, on their parameter space) [see our response to the next point, and Lemma 7]. The rest of our proposition’s results and proofs remain as before. We believe that this change strengthens the proposition by making its assumptions more realistic and more precise.
> >
> > Note that our results and proofs agree that “small loss does not imply small gradient”. We have added to the main text some clarification that the M smoothness constant can be large in practice, and the bound becomes vacuous near saddle points, which are key situations where small loss does not imply small gradient. Nevertheless, our mathematical result that the loss ratio upper bounds the gradient norm ratio is mathematically correct (under the assumptions), which shows that in some cases, a smaller loss ratio can imply a smaller gradient norm ratio. To assist the reader, we have removed overly strong language and added appropriate language to guard the interpretation of the proposition. Nevertheless, we believe that the proposition sheds light on the structure of the relationship between $\epsilon(\theta)$ and the gradients.
> >
> > We have also expanded our experimental study by training up to 3M training steps, extended 1M training steps. With this expansion, we now see statistically significant positive correlation at all flow matching times between the loss ratio and the gradient norm ratio, with pearson correlation ranging from 0.23 to 0.94.
> >
> > > The proof to Proposition 3 is based on several previous results and mathematical notions that are not defined in the paper, making it challenging to read and understand the proof. The author should at least define important concepts (like KL inequality), and prove all claims such as analyticity implies M-smoothn ess. I suspect these claims are not true in general, for example x^4 is analytic but not M-smooth on the real line.
> >
> > Thank you for bringing our attention to this. We agree that the previous version of proposition 3 was challenging to read and understand. We have amended its exposition in the main text and appendix, defining and contextualizing important concepts like the KL inequality. For ease of exposition, we provide it here:
> >
> > “The Kurdyka-Łojasiewicz (KŁ) inequality is a generalization of the Polyak-Lojasiewicz condition, itself a generalization of convexity, which has been used to study convergence rates of stochastic gradient descent in conditions that are more realistic to deep neural networks \citep{pmlr-v162-scaman22a}.
> > The Kurdyka-Łojasiewicz (KŁ) inequality holds locally under mild conditions, only requiring analyticity, and states that there exists a compact local neighborhood $U$ around any critical point $\theta^*$ and constants $c>0, \alpha \in [1,2)$ such that, for all $\theta \in U$:
> >
> > \begin{equation}
> >     \| \nabla f(\theta) \|^2 \geq c | f(\theta) - f(\theta^*)|^\alpha
> > \end{equation}
> >
> > This is mathematically an inequality in the opposite direction as smoothness. Whereas smoothness ensures the function does not change too quickly, the KŁ inequality says the function does not change too slowly, which is important for gradient descent convergence rates. For our purposes, smoothness gives an upper bound on the numerator, while the KŁ inequality provides a lower bound on the denominator. Combining the two gives the ratio bound.”
> >
> > We have added Lemma 7 to the appendix which proves that analyticity implies smoothness on compact subsets, which is explicitly the setting used in our propositions. Indeed, x^4 is analytic but not M-smooth on the real line, but the real line is not a compact subset of the domain of x^4; x^4 is M-smooth over compact subsets such as the interval [0,1].

---

> > > ### Author Response · Authors · 2025-10-23
> > >
> > > > Section 4.2, the decomposition here only makes sense when the equivariant models form a linear subspace. The author tries to follow Nordenfors et al in Appendix A.1. However, this claim only holds for linear models. The author claims that ``element-wise applied nonlinearities are equivariant”, but this is incorrect for most activation functions. For example, ReLU is not rotation equivariant.
> > >
> > > Thank you for bringing our attention to this. We have fixed the error in the appendix erroneously stating that element-wise nonlinearities are generally equivariant. We have added clarifications to the main text and appendix on the three assumptions that Nordenfors et al., 2025 relies on: (i) the symmetry group is compact and acts on finite-dimensional hidden spaces; (ii) the neural net non-linearities are equivariant; and (iii) the loss is invariant. We also added discussions in the main text and appendix on the applicability of each condition; a brief summary is that conditions (i) and (iii) are commonly satisfied without much thought, whereas condition (ii) on non-linearities is less automatic. We note that our propositions 4-6 are not impacted. Proposition 4 focuses on the linear force prediction head of EScAIP, where we verify that all three conditions are satisfied. In particular, there is no non-linearity in this head, so condition ii is bypassed. Propositions 5-6 explicitly operate on the assumed setting that Nordenfors’ framework applies (i.e., assumes that the three conditions are satisfied), and so is not impacted.
> > >
> > > > Minor issues:
> > > > Page 4: the author should (briefly) explain that the first term in the decomposition of the loss is averaged over T.
> > >
> > > Fixed
> > >
> > > > The M_\epsilon-smooth condition is never defined in the main text. There also seems to be a typo in its definition in the Proof of Proposition 2.
> > >
> > > We have added a definition of smoothness to the main text. For expositional ease, we have renamed M_\epsilon to M. We have fixed the typo.
> > >
> > > We have fixed the typos, updated Figure 2f’s legend ordering, and added text in figure 4c to distinguish prediction error and equivariance error.

---

> > > ### Comment · Reviewer_Rsqg · 2025-11-02
> > > **Thank you and some concerns about theoretical part**
> > >
> > > I would like to thank the authors for addressing most of my questions and expanded experiments in B.3. Overall I think the current manuscript is well written and has clear insights on training dynamics w.r.t equivariant loss.
> > >
> > > For the revised Proposition 2 and Proposition 3, the authors have made efforts to restrict the analysis over a compact domain surrounding the zeros of $\epsilon(\theta)$. However, I still think the claim "the loss ratio shrinks, the gradient
> > > norm ratio also shrinks." is not clearly supported by these propositions. The main reason is the analysis over domain $U$ would lead to a bound (in particular smoothness constant $M$) that depends on $U$. As a consequence, $M$ is also depending on $\epsilon(\theta)$. In some sense, the statement/bounds in these propositions can be misleading. I hope the authors can remark the dependence of $M$ over $\epsilon$ to avoid unnecessary confusion.
> > >
> > > In general the dynamics near $\epsilon$ can be quite challenging to investigate so it is not expected to provide explicit bound. I appreciate the experiment investigation below the propositions as they provides more insights along the training dynamics.

---

> > > > ### Author Response · Authors · 2025-11-03
> > > >
> > > > Thank you for your comment. We have revised the manuscript:
> > > >
> > > > - In propositions 2 and 3, we rewrite M-smoothness notation to M_f(U) for function f, on compact subset U, to explicitly notate the dependence of the smoothness constant on the function and the subset
> > > > - Reworded claim on experimental investigation: "Our propositions 2, 3 show that in certain conditions, a function of the loss
> > > > ratio upper bounds the gradient norm ratio."
> > > >
> > > > We believe these changes clarify the relationship more precisely and correctly, and help avoid unnecessary confusion.

---

### Decision · Action_Editor_asS6 · 2025-11-27

**Recommendation:** Accept as is

**Additional Comments:**

The authors have responded to the reviewers’ comments. The reviewers appreciated the answers and are of the opinion that the manuscript meets the TMLR acceptance criteria.

For the camera-ready version, please ensure that all requested modifications are incorporated. While the revisions have already addressed these points, kindly double-check to confirm that nothing remains outstanding.

**Audience:**

Yes

**Audience Explanation:**

The work addresses fundamental questions about how and how quickly neural networks learn symmetries, specifically 3D rotational equivariance, which is highly relevant to researchers in machine learning theory, geometric deep learning, and scientific applications such as molecular modeling.

**Claims And Evidence:**

Yes

**Claims Explanation:**

The paper introduces a principled metric for quantifying how much of a model’s error is due to imperfect equivariance, using a loss decomposition that is theoretically justified for any convex loss. Through extensive experiments on high-dimensional molecular tasks, the paper shows that standard neural networks trained with data augmentation rapidly learn 3D rotational equivariance, reducing equivariance error to less than 2% of total loss within the first 1,000–10,000 training steps, regardless of model or dataset size. Theoretical analysis further demonstrates that as equivariance error decreases, both the model’s gradients and parameters converge toward those of a perfectly equivariant model. These claims are supported by mathematical proofs, empirical results across multiple architectures and datasets, and clear acknowledgment of the method’s scope and limitations.

Please also look at the reviewer's comments.